# Hidden in the Haystack: Smaller Needles are More Difficult for LLMs to Find

## Abstract

Large language models (LLMs) face significant challenges with needle-in-a-haystack tasks, where relevant information ("the needle") must be drawn from a large pool of irrelevant context ("the haystack"). Previous studies have highlighted positional bias and distractor quantity as critical factors affecting model performance, yet the influence of *gold context size*, the length of the answer-containing document, has received little attention. We present the first systematic study of gold context size in long-context question answering, spanning three diverse benchmarks (general knowledge, biomedical reasoning, and mathematical reasoning), eleven state-of-the-art LLMs (including recent reasoning models), and more than 150K controlled runs. Our experiments reveal that LLM performance drops sharply when the gold context is shorter, i.e., **smaller gold contexts consistently degrade model performance and amplify positional sensitivity**, posing a major challenge for agentic systems that must integrate scattered, fine-grained information of *varying lengths*. This effect persists under rigorous confounder analysis: even after controlling for gold document position, answer token repetition, gold-to-distractor ratio, distractor volume, and domain specificity, gold context size remains a decisive, independent predictor of success. Our work provides clear insights to guide the design of robust, context-aware LLM-driven systems.

## 1 Introduction

Large language models (LLMs) increasingly power applications that require reasoning over vast amounts of information, such as synthesizing evidence across scientific literature (Sprueill et al., 2024; Bazgir et al., 2025; Wang et al., 2025; Gao et al., 2025) or navigating complex codebases (Liu et al., 2023; Zhang et al., 2023; Bogomolov et al., 2024).

A critical stage in such systems is *aggregation*, the synthesis of retrieved evidence into an accurate, actionable response. This stage determines what content to include, cite, or ignore, and has direct implications for safety, reliability, and factual correctness. One specific, common variant of aggregation is needle-in-a-haystack (NIAH) scenarios, where relevant evidence (the 'gold context') is embedded within a large volume of topically related or superficially plausible but ultimately irrelevant or misleading, 'distractor context' (Tay et al., 2021; Shaham et al.). Successful aggregation requires precise identification of essential evidence among a large number of distractor documents.

In NIAH scenarios, prior work has explored phenomena such as *positional bias* (Wang et al., 2023; Liu et al.), showing that models disproportionately attended to early content and that distractors degrade performance. Yet one key dimension remains underexplored: *how the size of the gold context influences model performance.* This question is critical for the design of agentic systems, where autonomous agents must perform NIAH-style aggregation over heterogeneous information streams produced by specialized components, whose evidence can *vary widely in length.*

In this study, we present a systematic analysis of gold context size as an independent variable in LLM performance in long-context, single-needle NIAH settings. Our main finding (§3) is that smaller gold contexts consistently (1) **degrade performance** and (2) **heighten sensitivity to positional bias**, as illustrated in Fig.1. This exposes a form of brittleness not captured in prior work.

To support our analysis, we adapt three diverse datasets spanning mathematics, biomedical reasoning, general knowledge, systematically varying both the size and position of the gold context *while*

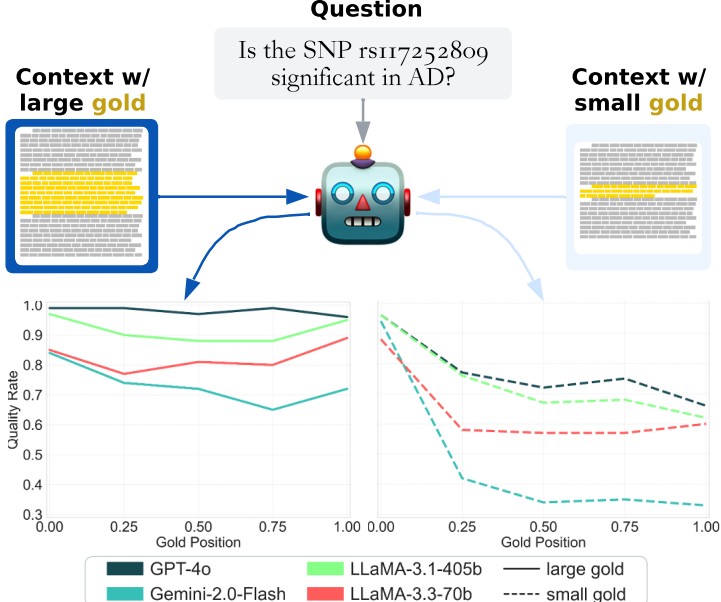

Figure 1: LLM performance on a needle-in-a-haystack task (CARDBiomedBench). The *y*-axis is a measure of accuracy (higher is better). Varying both the size and position of the gold context within fixed distractor content, we find that LLMs perform *worse* and exhibit *greater positional sensitivity* when given short gold documents (dashed line) compared to long ones (solid line).

*keeping the distractor content fixed* (§2). Our study comprises over 150K controlled runs across eleven state-of-the-art LLMs, both general-purpose and reasoning-focused, ensuring that the observed effects hold across architectures and tasks.

We conduct extensive confounder analyses to ensure that our findings are not driven by spurious factors (§4). Even after controlling for gold position, answer-token repetition, gold-to-distractor ratio, distractor volume, and domain specificity, gold context size remains a strong, independent predictor of performance. The overall pattern is consistent—larger gold contexts outperform smaller ones—though the magnitude of the effect varies across conditions. For instance, smaller gold contexts are particularly vulnerable to primacy bias, whereas larger gold ones exhibit greater robustness to both positional variation and distractor interference.

These findings have practical implications. In real-world deployments, factors such as context size, position, distractor length, and noise are typically uncontrolled. Our results show that heterogeneity in document length can induce aggregation failures, particularly when small but critical evidence is overshadowed by much longer passages. This situation naturally arises in retrieval-augmented and agentic systems, where an aggregator must isolate a single relevant piece of evidence from a mixture of heterogeneous sources. Practitioners can mitigate this fragility by monitoring length disparities among retrieved documents and by avoiding pipelines that combine extremely short and very long contexts, thereby reducing size-induced attention asymmetries.

In summary, our contributions are as follows:

- **Novel determinant of long-context performance:** To the best of our knowledge, we are the first to demonstrate that the *size of gold contexts* functions as a hidden variable in LLM performance on long-context NIAH. Smaller golds degrade accuracy and amplify positional bias, underscoring a potential fragility in real-world applications.
- **Robust to confounders:** We identify and analyze five potential confounding variables, (1) gold document position, (2) answer token repetition, (3) gold-to-distractor ratio, (4) distractor volume, and (5) domain specificity, demonstrating that gold context size remains a decisive predictor of success despite these factors.
- **Large-scale experimentation:** We repeat our experiments and aggregate findings across eleven state-of-the-art LLMs, three diverse benchmarks, three sizes of gold, and six positions in the context window totaling over 150k controlled runs.

## 2 EXPERIMENTAL SETUP

This section outlines our design objectives, benchmark adaptations, baseline validations.

### 2.1 DESIDERATA

We begin by outlining the core desiderata guiding our experimental design.

**Realism.** In real-world agentic systems, aggregation involves synthesizing outputs from multiple specialized agents, each retrieving information from their domain of expertise. Usually, one agent returns the document containing the correct answer (the "gold" document), while others provide distractors, topically relevant but ultimately uninformative. We simulate this by inserting a gold document of varying size at different positions within a fixed-length sequence of distractors. Document order is randomized to reflect natural uncertainty in agent contributions and retrieval quality.

**Gold Size Variability.** We constructed three nested gold variants for each benchmark:

- **Small Gold**: Minimal span sufficient to answer the question.
- **Medium Gold**: Additional explanatory or supporting content.
- **Large Gold**: Complete reasoning process and/or extended relevant context.

These were wrapped in pseudo-documents (titles, questions). Variants are hierarchically structured (small $\subset$ medium $\subset$ large) and validated for sufficiency. See Figure 9 for examples. Performance is high and uniform when observing only the gold of any size (Appendix B.1).

**Distractors.** To simulate realistic scenarios, we curate distractors topically relevant and lexically similar to the question but lacking the answer. The total distractor budget per benchmark is $\sim 20k$ tokens—a deliberate design choice that provides a sufficiently long distractor context to be considered 'long-context' while remaining computationally manageable for extensive experimentation.

**Generality.** We select three diverse benchmarks spanning biomedical, general knowledge, and mathematical reasoning, and evaluate performance across eleven leading LLMs of varying architecture and scale. This ensures that our findings generalize across domains and model classes.

### 2.2 TASK CONSTRUCTION: NEEDLES AND HAYSTACKS

We adapt three established question and answering benchmarks—CARDBiomedBench (biomedical reasoning), NaturalQuestions (general knowledge), and NuminaMath1.5 (mathematical reasoning)—to create controlled NIAH settings. Gold context sizes were varied, accompanied by distractors explicitly designed to be topically relevant yet answer-free. Figure 2 displays token count distributions for the varying sizes of gold. We provide further details on datasets and their metrics in §A.1.

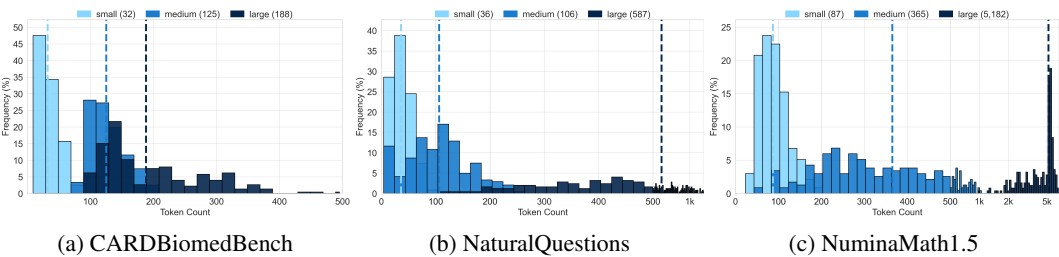

(a) CARDBiomedBench      (b) NaturalQuestions      (c) NuminaMath1.5

Figure 2: Token count distributions for varying sizes of gold context on each benchmark. Median token count is in parenthesis in the legend. X-axis is scaled as linear (0-500) and logarithmic (500+).

**CARDBiomedBench (CBB).** CBB is a question-answering dataset focused on neurodegenerative diseases, designed to evaluate LLM performance on biomedical reasoning tasks involving genetic, molecular, and clinical information. The BiomedSQL (for Alzheimer's & , CARD) variant augments each example with SQL queries and database rows to support structured reasoning experiments:

Full answer $\subset$ SQL query + answer $\subset$ SQL query + rows + answer

Distractors were drawn from documents retrieved by four independent domain-specific agents with access to biomedical knowledge bases. These documents are semantically relevant but verifiably do not contain the answer, presenting realistic aggregation challenges.

**NaturalQuestions (NQ).** NQ is an open-domain QA benchmark of Google queries, with Wikipedia passages from the KILT corpus as evidence (Petroni et al., 2021):

$$\boxed{\text{Sentence with the answer}} \subset \boxed{\text{Paragraph with the sentence}} \subset \boxed{\text{Paragraph} \pm 4 \text{ paragraphs}}$$

Distractors were drawn from the HELMET (Yen et al., 2025) adaptation of NQ-KILT as 100-token segments. We exclude any documents labeled as evidence or containing the answer. This ensures distractors remain lexically and topically aligned with the question, but free of the answer.

**NuminaMath1.5 (NM).** NM is the largest open-source dataset for math reasoning, with problems ranging from high school to International Mathematical Olympiad (IMO)-level difficulty, originating from diverse sources like Chinese K-12 exams, AMC/AIME contests, and global math forums. We used the OpenR1Math (R1, 2024) variant, which includes model-generated solution traces from DeepSeekR1 (DeepSeek-AI et al., 2025) verified for correctness. We filter for examples with complete reasoning streams and define gold variants as:

$$\boxed{\text{Full answer}} \subset \boxed{\text{Textbook-style solution + answer}} \subset \boxed{\text{Full LLM-generated chain-of-thought + solution + answer}}$$

Distractors were reasoning traces to different questions. Due to length variability, we cap large gold contexts at the final $5k$ tokens, which include concluding reasoning and answers.

### 2.3 BASELINE EXPERIMENTS

We run three baseline conditions to validate that observed performance differences in main experiments result from changes to gold size, rather than underlying flaws in datasets or distractor construction. **Baseline results across all benchmarks and models can be found in Appendix B.1**:

- **Closed-book.** No context is provided, assessing whether models could answer from internal knowledge. This gauges possible benchmark saturation.
- **Gold-only.** Each gold context is presented alone, without distractors. This confirms variants were sufficient to solve the task and that downstream performance drops are due to aggregation effects (e.g., distractor interference or gold placement).
- **Distractor-only.** Models are given only distractor documents. For CBB, we also test distractors from each agent separately to confirm they were individually non-informative. These checks ensure that distractors lack sufficient signal to answer correctly.

### 2.4 MAIN EXPERIMENTS

We simulate realistic aggregation scenarios by embedding each gold context size at varying positions within a fixed sequence of distractors. This tests both gold size and positional sensitivity simultaneously. We evaluate eleven leading LLMs:

- **Closed-weight**: o3-mini, GPT-4o (OpenAI et al., 2024), GPT-4o-Mini (OpenAI, 2024), Gemini-2.5-Flash, Gemini-2.0-Flash, and Gemini-2.0-Flash-Lite (Mallick & Kilpatrick, 2025)
- **Open-weight**: DeepSeek-R1 (DeepSeek-AI et al., 2025), Phi-4-reasoning (Abdin et al., 2025), LLaMA-3.1-405B, LLaMA-3.3-70B, and LLaMA-3.1-8B (Dubey et al., 2024)

We evaluate each model on every size-position combination in a deterministic setting. Prompts were standardized within each benchmark. This enables rigorous, cross-model evaluation of gold context sensitivity and simulates the type of unpredictable document ordering in LLM systems.

## 3 MAIN FINDING: SMALLER GOLD CONTEXTS LOWER THE PERFORMANCE

Our experiments reveal that gold context size has a substantial and consistent effect on long-context performance, irrespective of confounding variables, across different benchmarks and models.

Increasing the size of the gold context significantly improves accuracy (Figure 3). On CBB, Gemini-2.0-Flash went from 48% with small to 62% with medium and 73% with large. GPT-4o performs similarly, rising from 77% (small) to 98% (large), while LLaMA-3.1-405B went from 74% to 92%.

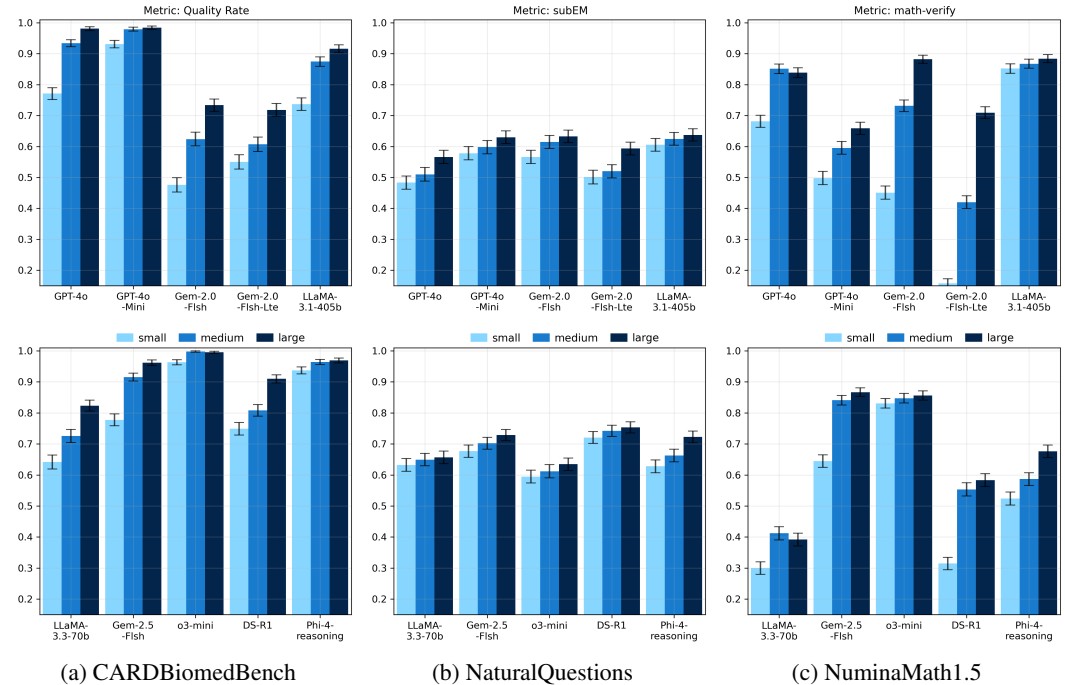

(a) CARDBiomedBench      (b) NaturalQuestions      (c) NuminaMath1.5

Figure 3: Average performance across all gold positions for each benchmark and gold context size. Metrics are benchmark-specific (BioScore, subEM, math-verify). Higher is better. Error bars indicate 90% confidence intervals. Colors correspond to gold context sizes: small, medium, large. **Across all settings, performance improves monotonically with gold context size.**

Notably, performance with large gold contexts approaches the *gold-only baselines* (i.e., accuracy when the gold context is shown without any distractors) recorded at 96% for Gemini-2.0-Flash, and 100% for both GPT-4o and LLaMA-3.1-405B. This suggests that large gold contexts allow models to nearly recover ideal aggregation performance, while small golds fall significantly short.

## 4 ANALYSIS OF CONFOUNDING FACTORS

Our goal is to isolate the effect of *gold context size* as an independent factor in LLM performance. However, there are various other factors that confounded or correlated with our target variable, making the attribution of the observed effect non-trivial. To better understand the underlying phenomenon here, one must control for any potential confounding factors that may impact the outcome of the findings. Specifically, in this section we study and control the following confounds: gold document position in the context window (§4.1), the total number of repetitions of the answer in the context (§4.2), relative length of gold document to the total distractor evidence length (§4.3), total distractor length (§4.4) and domain specificity (§4.5).

### 4.1 GOLD DOCUMENT POSITION

From the results in Figure 4, we observe that **smaller gold documents are hard to find *regardless of their position***. Nevertheless, **certain positions amplify the bias against smaller gold documents**. Performance systematically declines when small gold contexts appear later in the input, while large gold contexts are more robust to position (Full results in Appendix B.5).

For instance, in CBB, Gemini-2.0-Flash achieves 94% accuracy when the small gold context is placed at the start of the context window, but only 33% when placed near the end, a 61-point drop. In contrast, the large gold context declines more gradually, from 84% to 65%, demonstrating greater positional resilience. This pattern held across all evaluated models and benchmarks with some effects more amplified than others.

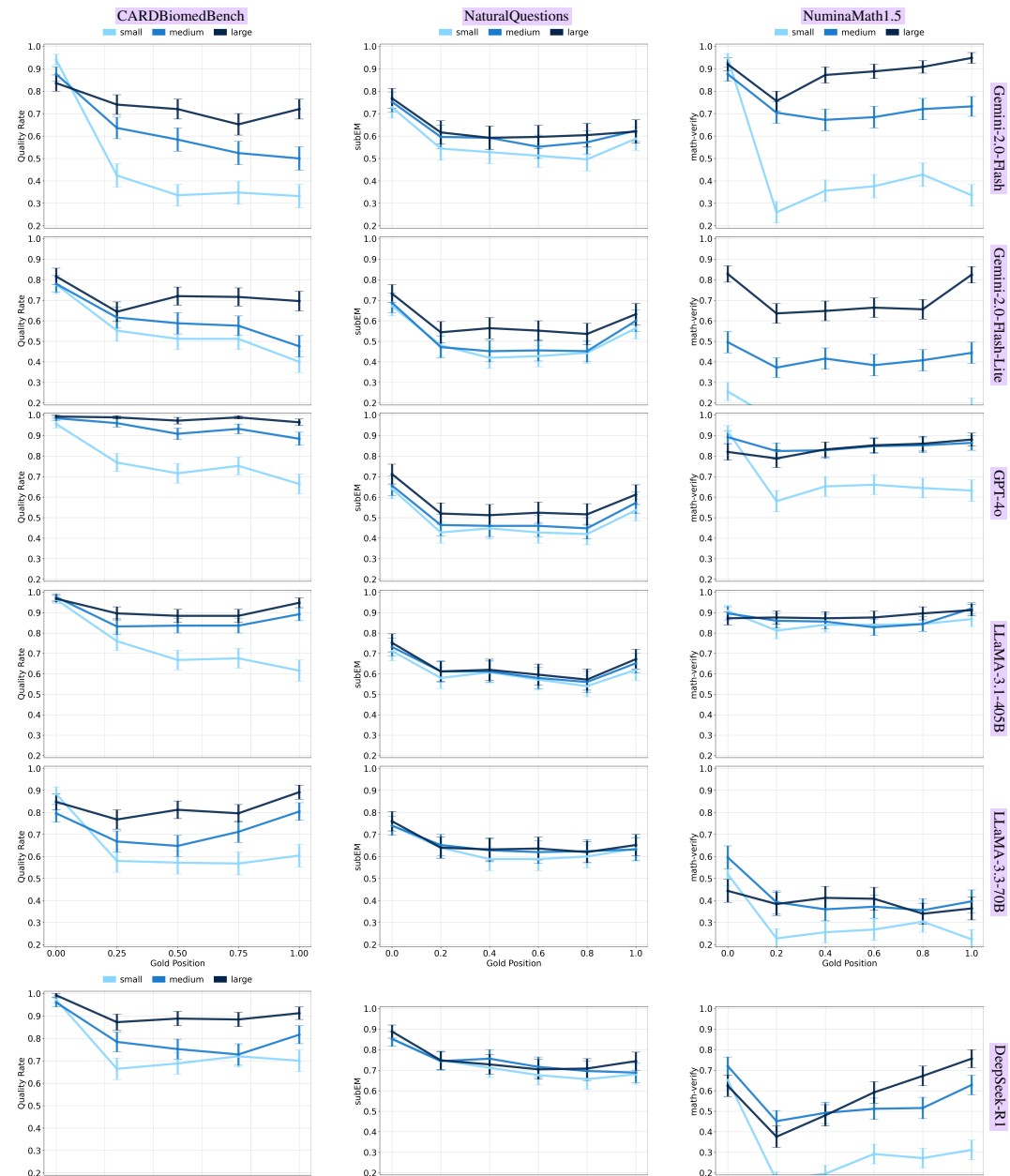

Figure 4: Model performance by gold context position (early to late in input), higher is better and error bars are 90% CIs. Each row is a model, columns are benchmarks. **Smaller gold contexts exhibit sharper performance degradation with later placement, especially in specialized domains (CBB, NM).** Larger contexts mitigate this sensitivity, highlighting the stabilizing effect of richer input.

Importantly, the positional effect is more pronounced in domain-specific tasks (CBB and NM) than in general knowledge (NQ), suggesting that task type and gold size compound aggregation difficulty.

We also observe that models **exhibit stronger primacy bias with smaller gold contexts**: performance is consistently higher when the gold context appears early in the input window. This effect is especially pronounced for small gold contexts. In some cases, small gold contexts placed at the beginning of the input even outperformed medium or large contexts placed later. This occurs often in the left and right columns of Figure 4, where the small gold line starts at the top at gold position 0.0 before crossing to the bottom.

This inversion highlights the sensitivity of model attention to positional cues when dealing with minimal evidence. While some bias exists for larger contexts, they are substantially more robust to position and do not exhibit the same sharp drop in middle and tail placements.

## 4.2 ANSWER TOKEN REPETITION

If larger gold documents contained the exact answer tokens more frequently, this could partly explain the phenomenon we observed—repeated encoding of the answer tokens would make it easier for the model to detect and attend to them. To test this, we compute *answer token repetition*, defined for an answer $a$ and a context $c$ as: $\mathrm{AnsTokRepetition}(a, c) = \frac{1}{|T(a)|^*} \sum_{t \in T(c)} \mathbf{1}[t \in T(a)]$, where $T(x)$ are the tokens of $x$, $|T(x)|^*$ is the number of *unique* tokens in $x$, and $\mathbf{1}[\cdot]$ is the indicator function. Distributions of this metric across gold sizes and benchmarks are provided in Appendix C.2. We then bucket tasks using the median repetition value (1.5) and compare performance within each bin for CBB (Figure 5).

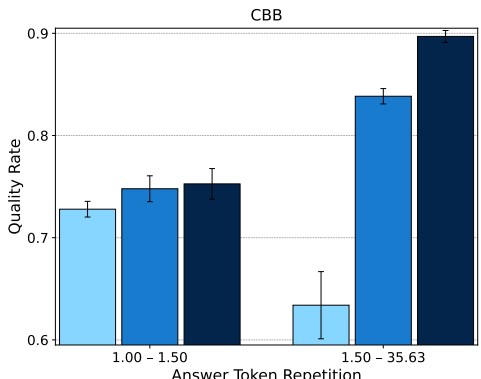

Figure 5: Performance across all models on CARDBiomedBench when bucketing tasks into two categories: answer token repetition less than or equal to, and greater than, 1.5. Smaller golds yield lower accuracy compared to larger golds across both bins. Error bars are 95% confidence intervals, with some wider due to sample size.

When answer repetition is low, larger gold contexts still outperform smaller ones, showing that repetition alone cannot explain the size effect. When repetition is high, performance improves for medium and large golds but not for small ones, indicating that repetition further amplifies the size effect. **While repetition is beneficial for LLM performance, it is insufficient to explain the observed size effect. Small gold contexts remain consistently harder to detect and use.**

## 4.3 GOLD-TO-DISTRACTOR RATIO

Given a fixed distractor budget, changing the size of the gold document also changes the proportion of gold tokens to distractor tokens within a context window. For a gold document $g$ and a distractor set $D$, we can compute: Gold-to-Distractor Ratio$(g, D) = \frac{|T(g)|}{\sum_{d \in D} |T(d)|}$, where $T(x)$ are the tokens of $x$ and $|T(x)|$ is the total number of tokens. This raises the question of, if the positive effect of larger gold documents is due to the size itself, or the increased ratio? Bucketing the tasks using the mean ratio (0.029) into similar ranges of gold-to-distractor ratio, we can see if size still has an effect when the ratio is held constant. Figure 6 shows just this, and larger golds consistently outperform smaller ones within bins. **Even after controlling for gold-to-distractor ratio, gold context size remains a strong indicator of performance.**

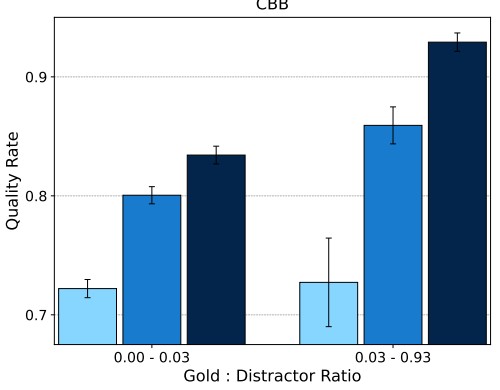

Figure 6: Performance across all models on CARDBiomedBench when bucketing tasks by the mean (0.029) for gold-to-distractor ratio. Smaller golds yields lower accuracy compared to larger golds. Error bars are 95% confidence intervals, some are larger due to small sample size in that bin.

## 4.4 DISTRACTOR VOLUME

To evaluate the robustness of the gold context size effect under varying degrees of context noise, we systematically increased the number of distractor documents. We leveraged our adaptation of NuminaMath1.5 to run experiments with 5, 10, and 15 distractors, approximately 25k, 50k, and 75k distractor tokens, respectively. Figure 7 shows that performance is strongly influenced by gold

context size, regardless of distractor volume. This reinforces that size remains a dominant variable, even when noise levels change.

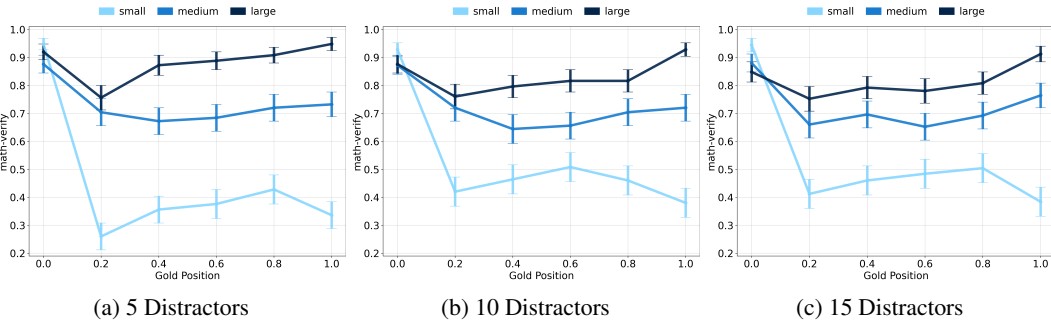

(a) 5 Distractors      (b) 10 Distractors      (c) 15 Distractors

Figure 7: Gemini-2.0-Flash performance on NuminaMath1.5 as the number of distractor documents increases (error bars are 90% CIs). Despite growing distractor noise (up to ∼75k tokens), the performance gap between small and large gold contexts persists. **This confirms that gold context size remains a key factor in long-context reasoning under high-noise conditions.**

## 4.5 DOMAIN SPECIFICITY OF TASKS

The effects of gold context size are notably amplified in domain-specific tasks compared to general knowledge. Figure 8 quantifies this for CBB by measuring the range in model performance across different gold context positions. (The results for other benchmarks are in Appendix B.6). For each model and gold size, we compute the performance range as the difference between maximum and minimum scores across all positions:

$$\text{Range} = \max_{i \in \{1,\dots,n\}} \text{perf}(\text{position}_i) \\ - \min_{i \in \{1,\dots,n\}} \text{perf}(\text{position}_i) \qquad (1)$$

For example, on NuminaMath1.5, Gemini-2.0-Flash showed a performance range of 72% for small gold contexts, compared to only 20% for large gold. A similar pattern held in CARD-BiomedBench. In contrast, NaturalQuestions exhibited smaller variation across all sizes,

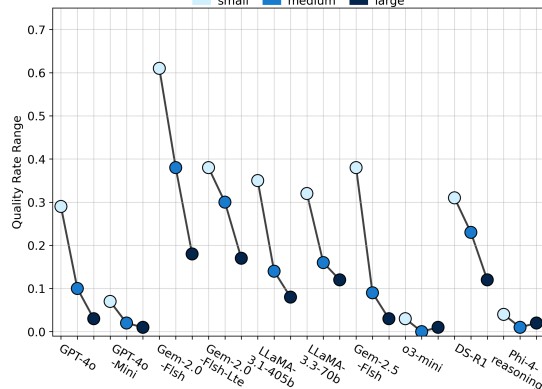

Figure 8: Positional sensitivity for CARDBiomed-Bench. For each model and gold context size, we compute the performance range across positions. **Smaller gold contexts exhibit much higher sensitivity (larger ranges), especially in domain-specific tasks. Larger gold contexts yield more stable performance across positions.**

likely due to easier questions and higher closed-book baseline scores. This suggests that general knowledge tasks may be inherently more resilient to gold context variability.

## 4.6 JOINT MODELING OF GOLD-SIZE & CONFOUNDERS WITH MULTIVARIATE REGRESSION

To assess the independent contribution of gold size relative to other factors, we fit a multivariate logistic regression predicting correctness from gold size, answer-token repetition, gold-to-distractor ratio, and position across all tasks and models. Earlier sections analyzed these variables *in isolation*, but in realistic long-context settings they co-vary, making it unclear which effects genuinely stem from gold size. The joint regression provides a unified *estimate of each factor's partial effect*.

**Model specification.** For each benchmark, we regress the binary correctness outcome on four categorical predictors:

$$\Pr(\text{correct}) = \sigma(\beta_0 + \beta_{\text{size}} X_{\text{size}} + \beta_{\text{rep}} X_{\text{rep}} + \beta_{\text{ratio}} X_{\text{ratio}} + \beta_{\text{pos}} X_{\text{pos}}) \qquad (2)$$

All variables are encoded categorically for direct interpretability. Repetition and ratio are binarized into *high* vs. *low* relative to within-benchmark means, and all predictors enter the model through their corresponding indicator variables $X_{\text{size}}, X_{\text{rep}}, X_{\text{ratio}}, X_{\text{pos}}$.

Here are how these variables defined: (1) **Gold Size ($X_{\text{size}}$).** A categorical factor encoding gold size, with *small* as the reference and indicators for *medium* and *large*. $\beta_{\text{size}}$ gives the change in log-odds when moving from *small* to larger contexts. (2) **Answer Token Repetition ($X_{\text{rep}}$).** Token overlap with the answer, binarized at the benchmark mean into *low* (reference) vs. *high*. $\beta_{\text{rep}}$ gives the log-odds difference between the two levels. (3) **Gold-to-Distractor Ratio ($X_{\text{ratio}}$).** The proportion of gold to distractor tokens in the context window, binarized at the benchmark mean into *low* (reference) vs. *high*. The coefficient $\beta_{\text{ratio}}$ quantifies the effect of moving from a lower share of gold tokens to a higher one. (4) **Position ($X_{\text{pos}}$).** The gold document's normalized start position, with $0.0$ as the reference. Indicators represent other positions, and $\beta_{\text{pos}}$ captures how placement in the window affects correctness.

**Results.** We show CBB results in Table 1 where we see that gold size has a strong independent effect: medium and large contexts significantly boost correctness after adjusting for all confounders. Repetition and ratio also help, while position has the largest negative impact—performance drops sharply when the gold appears anywhere other than at the start of the window. **CBB performance improves reliably with larger gold contexts, in ways not explained by repetition, ratio, or position alone.**

Appendix C.3 also reports all three benchmarks. **Across all benchmarks, gold context size remains a significant, independent predictor of correctness even after controlling for repetition, ratio, and position.** Medium and large golds consistently increase the log-odds of producing the correct answer relative to small ones.

| Predictor | Coef | 95% CI |
|---|---|---|
| Intercept | 1.6459 | [1.571, 1.720] |
| *Gold Size* | | |
| md vs. sm | 0.3933 | [0.336, 0.451] |
| lg vs. sm | 0.5354 | [0.466, 0.605] |
| *Answer Token Repetition* | | |
| high vs. low | 0.5924 | [0.511, 0.674] |
| *Gold-to-Distractor Ratio* | | |
| high vs. low | 0.3699 | [0.279, 0.461] |
| *Position* | | |
| 0.25 vs. 0.0 | -0.8087 | [-0.896, -0.721] |
| 0.50 vs. 0.0 | -0.9001 | [-0.987, -0.813] |
| 0.75 vs. 0.0 | -0.8930 | [-0.980, -0.806] |
| 1.00 vs. 0.0 | -0.8701 | [-0.957, -0.783] |

Table 1: Logistic Regression Results (CBB)

## 5 RELATED WORK

We review related work in the context of long-context reasoning, focusing on three themes: positional biases in LLMs, long-context evaluation frameworks, and mitigation strategies.

**Long-context biases in LLMs.** Position bias, the tendency of LLMs to over- or under-attend to different parts of the input, has emerged as a fundamental challenge. Prior work has identified several variants: *primacy bias*, where early content is favored (Wang et al., 2023); *recency bias*, where later content dominates (Zheng et al., 2023); and *U-shaped bias*, where mid-context is under-attended (Liu et al.). These effects persist across model architectures, alignment strategies (Liu et al.), extended context lengths (Lee et al., 2024; Veseli et al., 2025), and, to some extent, in internal representations (Lu et al., 2024). Our work contributes to this literature by introducing a new dimension: we show that *the size of the gold context modulates the strength of positional bias*.

A few recent works have examined different trade-offs in NIAH settings. Levy et al. (2024) study variable input lengths and show that, for a *fixed* gold document, *adding more distractors* degrades performance. This is essentially the inverse of our setup, where we fix the distractor content and systematically vary the length of the gold context. The parallel work of Levy et al. (2025) investigates the case where *all documents have equal length* while *the total context length is held fixed*. They find that *longer documents* (with fewer total documents) make the gold easier to discover. This is complementary to our work: instead of varying absolute length under equal-length constraints, we keep distractors fixed and vary *the relative size of the same gold evidence*. Finally, Dai et al. (2024) study NIAH performance in synthetic key–value retrieval tasks, focusing on varying answer-span lengths and task lengths. In contrast, we *hold the question and answer fixed* and vary the size of the surrounding gold context in open-ended QA, enabling a controlled analysis of how gold context size alone affects aggregation robustness.

**Frameworks for long-context evaluation.** Long-context evaluation has progressed from synthetic toy tasks to increasingly realistic settings. Early work such as Long-Range Arena (Tay et al., 2021) introduced standardized tasks for comparing transformer variants. Subsequent benchmarks expanded this space (Bai et al.; Li et al., a; Gao et al.; Li et al., b; Modarressi et al., 2025; Ling et al., 2025; Zhang et al.; Ye et al., 2024; Yen et al., 2024), exploring document synthesis (Shaham et al.; 2023), document-level retrieval (Yen et al., 2025), citation verification (Zhang et al., 2024a), and biomedical reasoning (Adams et al., 2024; Cui et al., 2025). Most adopt a needle-in-a-haystack (NIAH) formulation (Kamradt, 2023; Hsieh et al., 2024a), where a small relevant span must be recovered from distractors. Others move beyond strict NIAH by incorporating aggregation, multi-hop inference (Zhuang et al.; Katsis et al.), or mixed-modality inputs (Wu et al.). Our work builds on this trajectory by adapting natural, domain-specific datasets to simulate realistic NIAH aggregation.

**Mitigation strategies for long-context biases.** Prior work proposes various methods for reducing positional sensitivity, including context compression (Jiang et al., 2024), distilling long-context information into model weights (Cao et al., 2025), attention calibration (Hsieh et al., 2024b), modified positional encodings (Zhang et al., 2024b; Zheng et al., 2024), and debiased fine-tuning (Xiong et al., 2024). While these methods can alleviate positional bias, many introduce side effects (Zhao et al., 2024), and robust long-context generalization remains challenging. Our contribution is diagnostic rather than corrective: we uncover a novel bias connected to the gold context size. Whether existing mitigation strategies can address this effect remains an open question.

## 6 DISCUSSION, LIMITATIONS, AND CONCLUSION

**Why does gold context size strongly affect the accuracy?** At a high level, the finding here is pretty intuitive: Larger gold contexts spread semantically relevant information across more tokens, making the signal more resilient to positional noise and less likely to be overlooked amid distractors. However, we do not claim this is the only mechanism involved. Our findings suggest that interacting factors may be at play here, as discussed in §4—including positional dynamics, answer-context richness, evidence competition, and domain effects. While the overall trend (large > medium > small) is consistent across all analyses, the precise interaction among these factors remains open and is an important direction for future work.

**Practical implications of our findings.** While prior work has studied factors like positional bias and distractor count, our results highlight an overlooked factor: the effect of heterogeneity in the size of evidence documents. Aggregation quality can degrade sharply when small-but-critical evidence is mixed with much longer retrieved passages. This naturally arises in retrieval-augmented systems or agentic pipelines where an aggregator must select a single piece of evidence from a concatenation of heterogeneous sources. Practitioners may mitigate this by ensuring that the documents are balanced in lengths before being aggregated or by avoiding pipelines that combine extremely short and very long contexts, thereby reducing size-induced attention asymmetries.

**Limitations of our study.** We fixed distractor lengths to better reflect real-world conditions, resulting in varying gold-to-distractor ratios. This may confound whether performance differences stem from gold context size alone or its relative share. Parallel work has investigated this, offering complementary results to ours (Levy et al., 2025). Second, the tasks we use could be further grounded in real deployment scenarios. Third, our study focuses on aggregating a single needle; extending this to multi-needle or multi-hop settings is an important next step. Future work should address these.

**Conclusion.** Our study reveals a fundamental yet previously overlooked limitation in LLM aggregation capabilities: **the size of relevant information critically influences aggregation effectiveness in long-context tasks**. Through systematic evaluation, we demonstrated that smaller gold contexts degrade model performance substantially and exacerbate positional sensitivity, especially in domain-specific tasks. This discovery underscores a crucial vulnerability in real-world agentic deployments, where relevant evidence often appears unpredictably scattered amidst extensive distractors. As language models become central to applications requiring precise and trustworthy reasoning-from scientific discovery to personalized assistants-our findings highlight the urgent need to rethink aggregation strategies. Future LLM-driven systems must explicitly address context-size variability to ensure reliability, safety, and user trust in the face of complex, noisy real-world information streams.

# 7 REPRODUCIBILITY STATEMENT

In the Supplementary Material, we have included a folder with our self-contained code and instructions to reproduce all of the experimental results from this paper. Upon publication, we plan to push the code to a public GitHub repository, along with the data used to run the experiments, in order to support reproducible research.

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

# A  APPENDIX

## A  EXTENDED EXPERIMENTAL DETAILS

We provide extended experimental details on benchmark construction, model configuration, and evaluation methodology to support the reproducibility and interpretability of our results.

### A.1  ORIGINAL BENCHMARKS

We describe the sources, licenses, and preprocessing procedures for each of the three adapted benchmarks used in our experiments. All experiments were run on a sampled subset of 250 examples per benchmark. See the code repository for exact methodology.

**CARDBiomedBench**

- **Source:** CARDBiomedBench on Hugging Face and its BiomedSQL variant on Hugging Face. Distractor documents were retrieved using a multi-agent retrieval system, which retrieves content from: (1) Google search over NIH domains, (2) PubTator3.0 (Wei et al., 2024), (3) the Human Gene Mutation Database (HGMD) (Stenson et al., 2020), and (4) NCBI gene and variant pages (National Center for Biotechnology Information (NCBI), 1988).

- **License:** Apache 2.0 for benchmark code and data. Some distractor sources (e.g., HGMD) are not redistributable but are publicly accessible on their respective platforms.

- **Preprocessing:** None, the distractor and gold documents are as-is from the retriever.

**NaturalQuestions**

- **Source:** NQ with evidence spans aligned to Knowledge Intensive Language Tasks (KILT) on Hugging Face. Gold documents were loaded using the Ai2 ir_datasets python package (MacAvaney et al., 2021) and distractors were sourced from HELMET on Hugging Face.

- **License:** Creative Commons Share-Alike 3.0 (NQ), MIT (KILT & HELMET), and Apache 2.0 (ir_datasets).

- **Preprocessing:** We filtered for validation examples that had matching HELMET distractors. Examples with missing KILT provenance, absent or unresolvable answer spans, or malformed metadata were excluded. Gold and distractor documents included the title of the article *'Title: {title} Document: {gold_document}'* to give them context.

**NuminaMath1.5**

- **Source:** NuminaMath1.5 (NM) and its OpenR1Math (OR1M) variant on Hugging Face, that contains DeepSeekR1 reasoning chains.

- **License:** Apache 2.0. (NM and OR1M).

- **Preprocessing:** Filtered to retain only examples with 'complete' and 'verified' fields for question, final answer, structured solution, and long-form generation. DeepSeekR1 generations were truncated to the final 5,000 tokens using GPT-4o tiktoken (OpenAI, 2023) tokenization to normalize document length across tasks. Distractors sampling was among the other questions and excluded duplicates. Sizes of gold and distractors were strung into a pseudo-document by including *'The answer to {question} is {gold_document}'* to give them context.

## A.2 TASK CREATION

Here we provide more context for the gold document construction as discussed in §2.2. Specifically, in Figure 9 we show examples of the gold answers, of different sizes, for each dataset.

| | CARDBiomedBench (CBB) | NaturalQuestions (NQ) | NuminaMath1.5 (NM) |
|---|---|---|---|
| **Question** | What is the genomic location of rs12255438 in the GRCh38/hg38 build of the human genome and what gene is it located on or near? | Who is playing the halftime show at super bowl 2016? | A ship traveling along a river has covered 24 km upstream and 28 km downstream... Determine the speed of the ship in still water and the speed of the river. |
| **Small Gold** | The SNP rs12255438 is located on or closest to the gene CTNNA3 on chromosome 10 at base pair position 66465707 in the GRCh38/hg38 build of the human genome. | Super Bowl 50 halftime show It was headlined by the British rock group Coldplay with special guest performers Beyoncé and Bruno Mars, who previously had headlined the Super Bowl XLVII and Super Bowl XLVIII halftime shows, respectively. | The answer to the question "A ship traveling along a river has covered 24 km ..." is: v_{R}=4\mathrm{-}/\mathrm{}, v_{B}=10\mathrm{-}/\mathrm{} |
| **Medium Gold** | SELECT 'AlzheimerDisease_GeneData' AS source_table, UUID, SNP, chr_38, bp_38, nearestGene ... WHERE SNP = 'rs12255438' LIMIT 100 The SNP rs12255438 is located on or closest to the gene CTNNA3 on chromosome 10 at base pair position 66465707 in the GRCh38/hg38 build of the human genome. | Super Bowl 50 halftime show The Super Bowl 50 Halftime Show took place on February 7, 2016, at Levi's Stadium in Santa Clara, California as part of Super Bowl 50. It was headlined by the British rock group Coldplay with special guest performers Beyoncé and Bruno Mars, who previously had headlined the Super Bowl XLVII and Super Bowl XLVIII halftime shows, respectively. | Let $t$ be the time required for the boat to travel $24 \mathrm{-km}$ upstream and $28 \mathrm{-km}$ downstream, $v_{R}$ the speed of the river, and $v_{B}$ the speed of the boat. When the boat is traveling upstream, its speed is $v_{B}-v_{R}$, and when it is traveling downstream, its speed is $v_{B}+v_{R}$. ... The speed of the river is $v_{R}=4 \mathrm{-km} / \mathrm{h}$, and the speed of the boat is $v_{B}=10 \mathrm{-km} / \mathrm{h}$. |
| **Large Gold** | SELECT 'AlzheimerDisease_GeneData' AS source_table, UUID, SNP, chr_38, bp_38, nearestGene ... WHERE SNP = 'rs12255438' LIMIT 100 [{'SNP': 'rs12255438', 'chr_38': 10, 'bp_38': 66465707, 'nearestGene': 'CTNNA3'}, ... {'SNP': 'rs12255438', 'chr_38': 10, 'bp_38': 66465707, 'nearestGene': 'CTNNA3'}] The SNP rs12255438 is located on or closest to the gene CTNNA3 on chromosome 10 at base pair position 66465707 in the GRCh38/hg38 build of the human genome. | Super Bowl 50 halftime show The Super Bowl 50 Halftime Show took place on February 7, 2016, at Levi's Stadium in Santa Clara, California as part of Super Bowl 50. It was headlined by the British rock group Coldplay with special guest performers Beyoncé and Bruno Mars, who previously had headlined the Super Bowl XLVII and Super Bowl XLVIII halftime shows, respectively. ... At that time, Mars and Beyoncé were both doing a diet and stressing out. One day before the performance they were "watching playback backstage", while Beyonce ate a bag of Cheetos. ... (+5 more paragraphs) | <think> Okay, so I need to find the speed of the ship in still water and the speed of the river. Let me start by recalling that when a ship is moving upstream, its effective speed is the speed of the ship minus the speed of the river. ... Wait, actually, the problem states: "For this journey, it took half an hour less than for traveling 30 km upstream ...Hmm, let me parse that again... ... the final answer is v_{R}=4\mathrm{-}/\mathrm{},v_{B}=10\mathrm{-}/\mathrm{} |

Figure 9: Gold context construction across benchmarks. The "small" gold context is minimally sufficient to answer the question; "medium" and "large" add further relevant information. In CARDBiomedBench (left), this includes SQL and result rows; in NQ (center), adjacent Wikipedia paragraphs; in NM (right), full solution traces and DeepSeekR1 reasoning chain.

## A.3 LLM CONFIGURATION

We evaluated seven LLMs, each configured via provider-specific APIs. All evaluations were conducted as deterministically as possible.

**API Providers.** We used the following service providers for model access:

- **GPT models** (o3-mini, GPT-4o, GPT-4o-mini) were accessed via the `Azure OpenAI` service.
- **Gemini models** (Gemini-2.5-Flash, Gemini-2.0-Flash, Gemini-2.0-Flash-Lite) were accessed via the `Google AI GenAI SDK`, using the official `genai` Python client.
- **DeepSeek-R1 and Phi-4-reasoning** were accessed via the `Azure AI Inference` service.
- **LLaMA models >= 70b params** (Meta-LLaMA-3.1-405B-Instruct, LLaMA-3.3-70B-Instruct) were accessed via the `Azure AI Inference` service.
- **LLaMA model < 70B parameters** (LLaMA-3.1-8B-Instruct) was evaluated locally using the `meta-llama/Llama-3.1-8B-Instruct` checkpoint, loaded via Hugging Face `transformers`. All local evaluations were conducted on the NIH High-Performance Computing (HPC) Biowulf cluster (NIH Biowulf, 2024), leveraging GPU nodes for inference.

**Prompting and Evaluation Configuration.** Prompts were benchmark-specific and standardized across model types. All non-reasoning models were queried with `max_tokens=256` and

`temperature=0.0`. Provider-specific configurations (e.g., safety settings for Google GenAI, and device mapping for HuggingFace) were handled automatically during model initialization. See the code and YAML config files for full details. Reasoning models were queried with `max_tokens=2048` and their default generation params, to allow for reasoning. Reasoning models were additionally given instructions to encourage grounding their answer in the retrieved documents, to prevent relying on internal knowledge.

**Grading LLMs.** For CARDBiomedBench, an additional grading LLM was used to assess answer correctness via BioScore using GPT-4o, as done by the authors. It was instantiated using the same infrastructure and configurations as the primary LLMs, with `max_tokens=10`.

## A.4 METRICS

We used evaluation metrics that align with the original datasets' scoring protocols:

**Quality Rate.** We evaluate responses to the CBB tasks following their proposed **BioScore** framework, an LLM-as-a-judge metric implemented with GPT-4o. Each response is scored on a 3-point scale according to the BioScore prompt 4, and a score $\geq 2$ is considered factually correct. The **Quality Rate** is computed as the proportion of responses meeting this threshold.

Formally, given a reference set $Resp$ of expert-annotated responses and a corresponding set $\hat{Resp}$ of model-generated responses for $n$ questions:

$$\text{Quality Rate} = \frac{1}{N} \sum_{n=1}^{N} Correct(r_n, \hat{r}_n) \tag{3}$$

$$\text{where} \quad Correct(r_n, \hat{r}_n) = \begin{cases} 1, & \text{if} \quad \text{BioScore}(r_n, \hat{r}_n) \geq 2 \\ 0, & \text{otherwise} \end{cases} \quad \text{and} \quad r_n \in Resp, \hat{r}_n \in \hat{Resp} \tag{4}$$

**SubEM.** For NQ we utilized substring exact match, which assigns a score of 1.0 if any normalized ground truth string is a subspan of the model's response (after normalization), and 0 otherwise. This is a correctness signal used by previous work on this data.

**math-verify.** Evaluated with math-verify (Face, 2025), a symbolic equivalence checker that parses LaTeX boxed answers and verifies correctness through structured math expression comparison. Parsing and verification are done using an extraction and comparison pipeline derived from the Math-Verify toolkit.

**Error Bars.** All plots showing aggregate scores (e.g., Figure 3) report 90% confidence intervals (CIs) estimated via non-parametric bootstrapping over tasks. Given $N$ scores, we resample with replacement 1,000 times and compute the middle 90% interval from the resulting bootstrap distribution.

## A.5 PROMPTS.

We show prompts used to collect results from the models and the BioScore grading prompt. There is a unique prompt for each benchmark, which is used on every model. {Variables} are in curly braces which are formatted with task data (question and documents). We encourage models to ground their answers in the context and abstain if unable to answer.

```
You are a highly knowledgeable and experienced expert in the healthcare and biomedical field,
possessing extensive medical knowledge and practical expertise. Create an answer to the question
using only the provided documents (some of which might be irrelevant). If you cannot answer the
question based on the documents, explicitly state that you do not know.
Question: {question}
Documents: {documents}
```

Prompt 1: The CARDBiomedBench prompt is adapted from the original paper's experimental methods and includes mention of biomedical expertise.

```
Create an answer to the question using only the provided documents (some of which might be
irrelevant). If you cannot answer the question based on the documents, explicitly state that
you do not know.
Question: {question}
Documents: {documents}
```

Prompt 2: The NaturalQuestions prompt is adapted from previous work's experimental methods (Liu et al.; Yen et al., 2025).

```
Create an ANSWER to the QUESTION using only the provided DOCUMENTS (some of which might be
irrelevant). Write nothing but your final answer in LaTeX within \\boxed{}. If you do not
know the answer to a question, explicitly state so in \\boxed{I don't know}.
QUESTION: {question}
DOCUMENTS: {documents}
QUESTION: {question}
ANSWER:
```

Prompt 3: The NuminaMath1.5 prompt uniquely repeats the question and has formatting guidelines, to comply with the math-verify metric. Without repeating the question models exhibited extremely poor performance in every configuration.

```
You are a highly knowledgeable and experienced expert in the healthcare and biomedical field, possessing
extensive medical knowledge and practical expertise.
### Scoring Instructions for Evaluating Analyst Responses

**Objective:** Evaluate an analyst's response against a gold standard.

**Scoring Criteria:**
    - **Exact Match:** 3 points for an exact or equally accurate response.
    - **Close Match:** 2 points for a very close response with minor inaccuracies.
    - **Partial Match:** 1 point for a partially accurate response with significant omissions.
    - **Irrelevant Information (Harmless):** Deduct 0.5 points for harmless irrelevant information.
    - **Irrelevant Information (Distracting):** Deduct 1 point for distracting irrelevant information.
    - **No Match:** 0 points for no match.
    - **Not Knowing Response:** -1 point for stating lack of knowledge or abstaining. An example
    of this scenario is when Analyst Response says \'There are various studies, resources or
    databases on this topic that you can check ... but I do not have enough information on this topic.

**Scoring Process:**
    1. **Maximum Score:** 3 points per question.
    2. **Calculate Score:** Apply criteria to evaluate the response.

**Question:** {question}
**Golden Answer:** {gold_ans}
**Analyst Response:** {pred_ans}

## Your grading
Using the scoring instructions above, grade the Analyst Response return only the numeric score
on a scale from 0.0-3.0. If the response is stating lack of knowledge or abstaining, give it
-1.0.
```

Prompt 4: BioScore grading prompt for LLM-as-a-judge on CBB tasks, awarding points for correct information and deducting points for incorrect information. It differentiates an abstention (-1) from an incorrect answer (0 or 1).

## B    EXTENDED RESULTS

As discussed in §2.3 we evaluated *gold-only* baselines where each gold context size (small, medium, large) was presented alone, without distractors. This verified that all variants were independently sufficient to solve the task and that downstream performance drops are due to aggregation effects (e.g., distractor interference or gold placement). We provide baselines for all benchmarks in B.1 Figure 10.   Additionally, we provide full results for CBB in Figure 11, for NQ in Figure 12, and for NM in Figure 13. Finally, we provide positional curves for all models in Figure 15.

## B.1 BASELINES

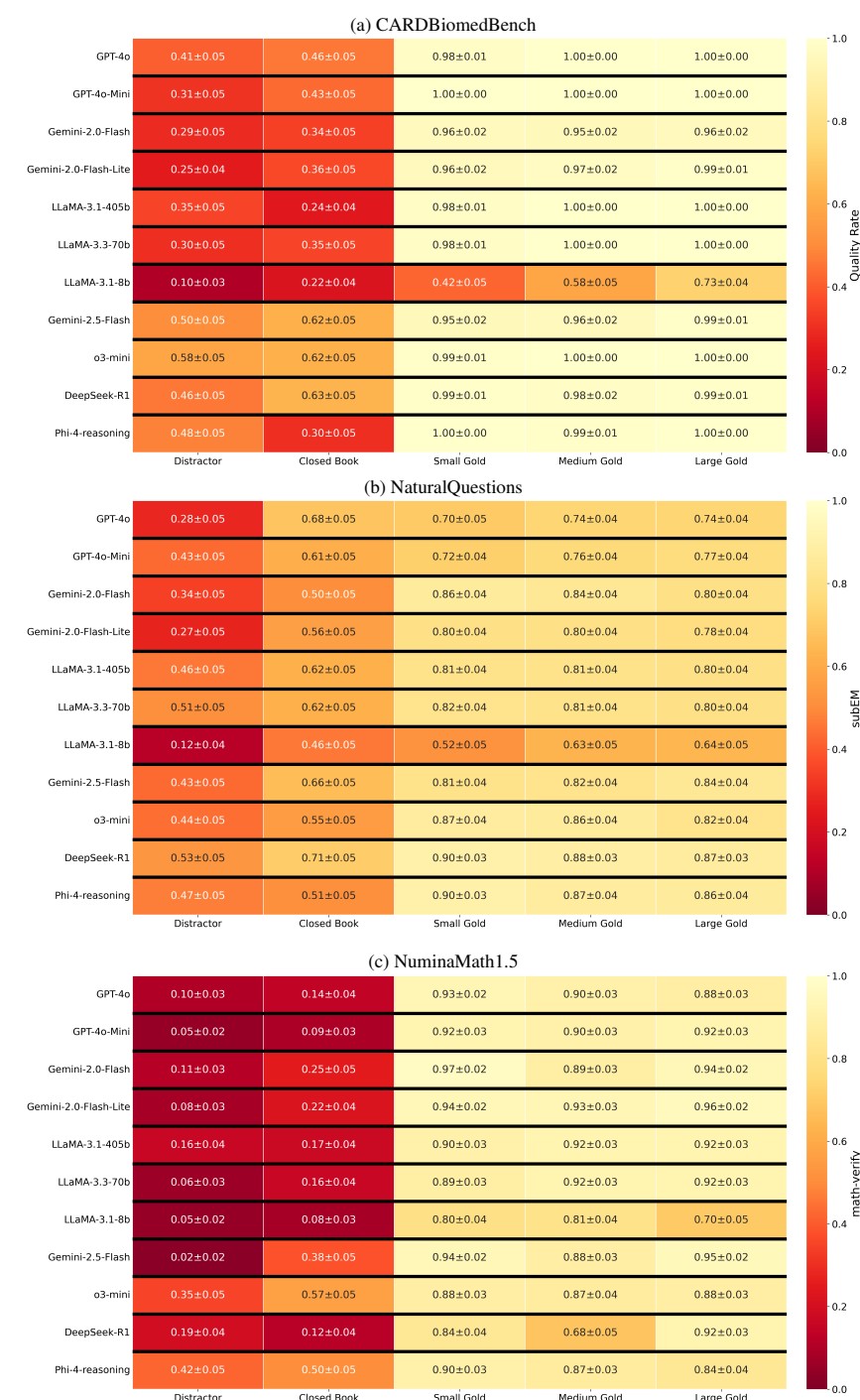

Figure 10: Baseline performance when viewing distractors only, closed book (no documents), and varying sizes of gold. This confirms both (1) performance is near perfect when viewing any size of gold document and (2) models perform poorly without the gold documents.

## B.2 CARDBIOMEDBENCH

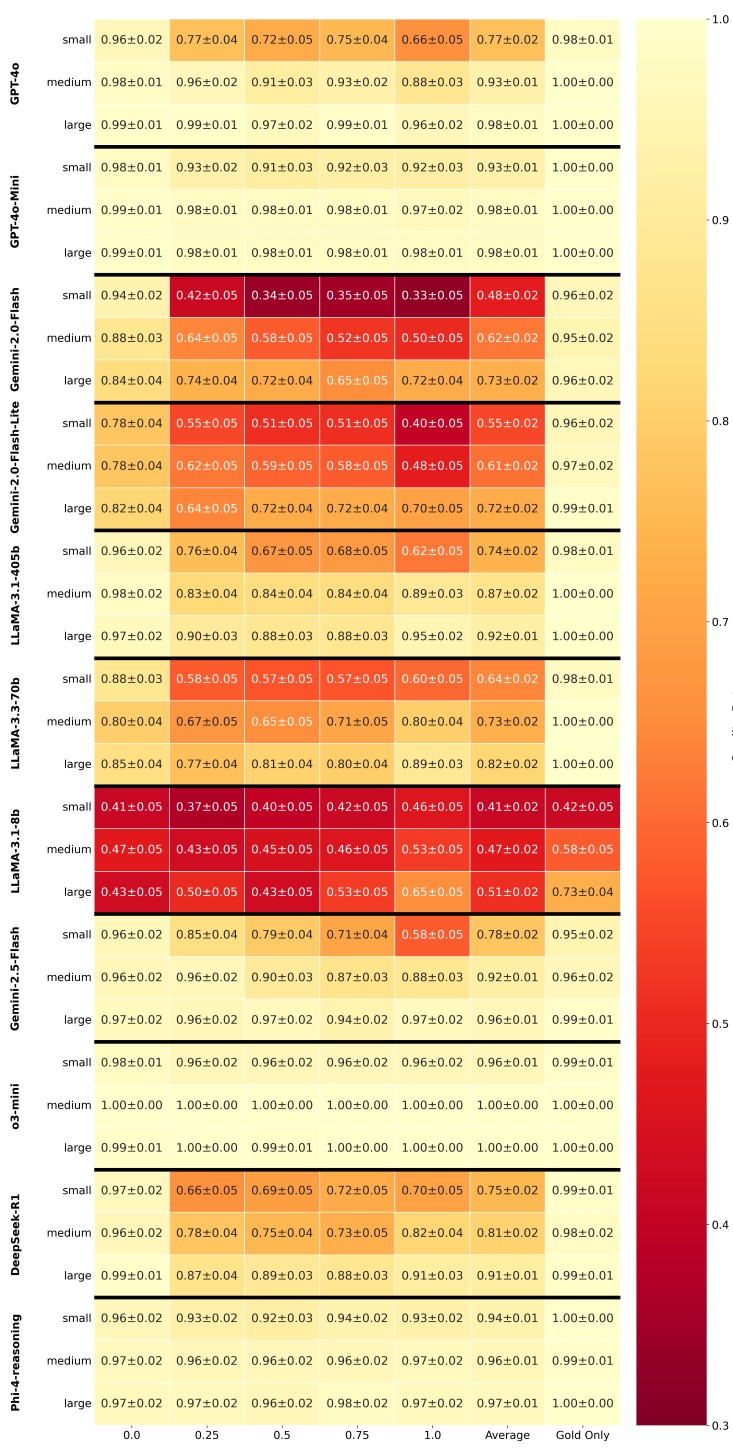

Figure 11: CARDBiomedBench performance for each model and size of gold for varying positions in the context window (0.0, 0.25, 0.5, 0.75, 1.0), the average across all positions, and baseline performance when seeing gold only. Higher scores (light yellow) is more desirable than low scores (dark red), 90% CI are reported.

## B.3 NATURALQUESTIONS

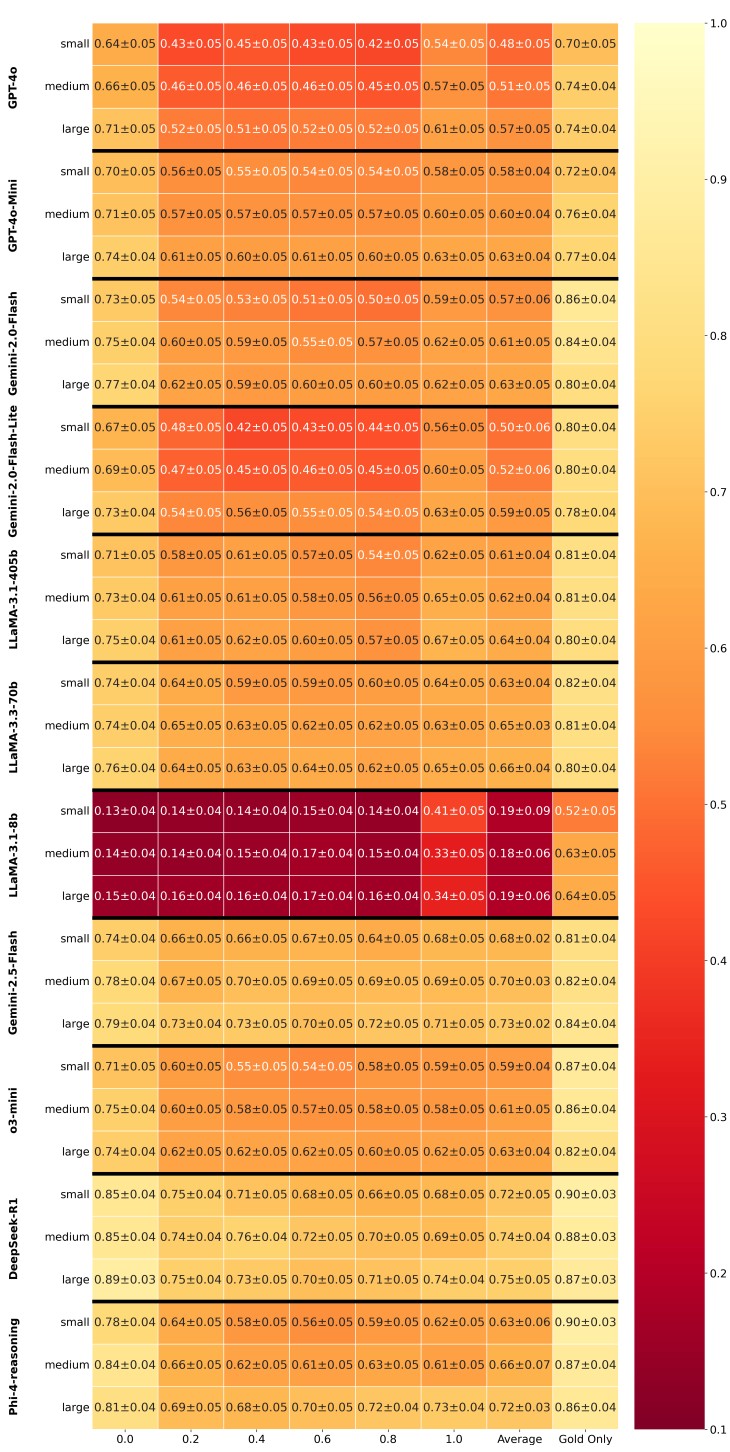

Figure 12: NaturalQuestions performance for each model and size of gold for varying positions in the context window (0.0, 0.2, 0.4, 0.6, 0.8, 1.0), the average across all positions, and baseline performance when seeing gold only. Higher scores (light yellow) are more desirable than low scores (dark red), 90% CI are reported.

## B.4 NUMINAMATH1.5

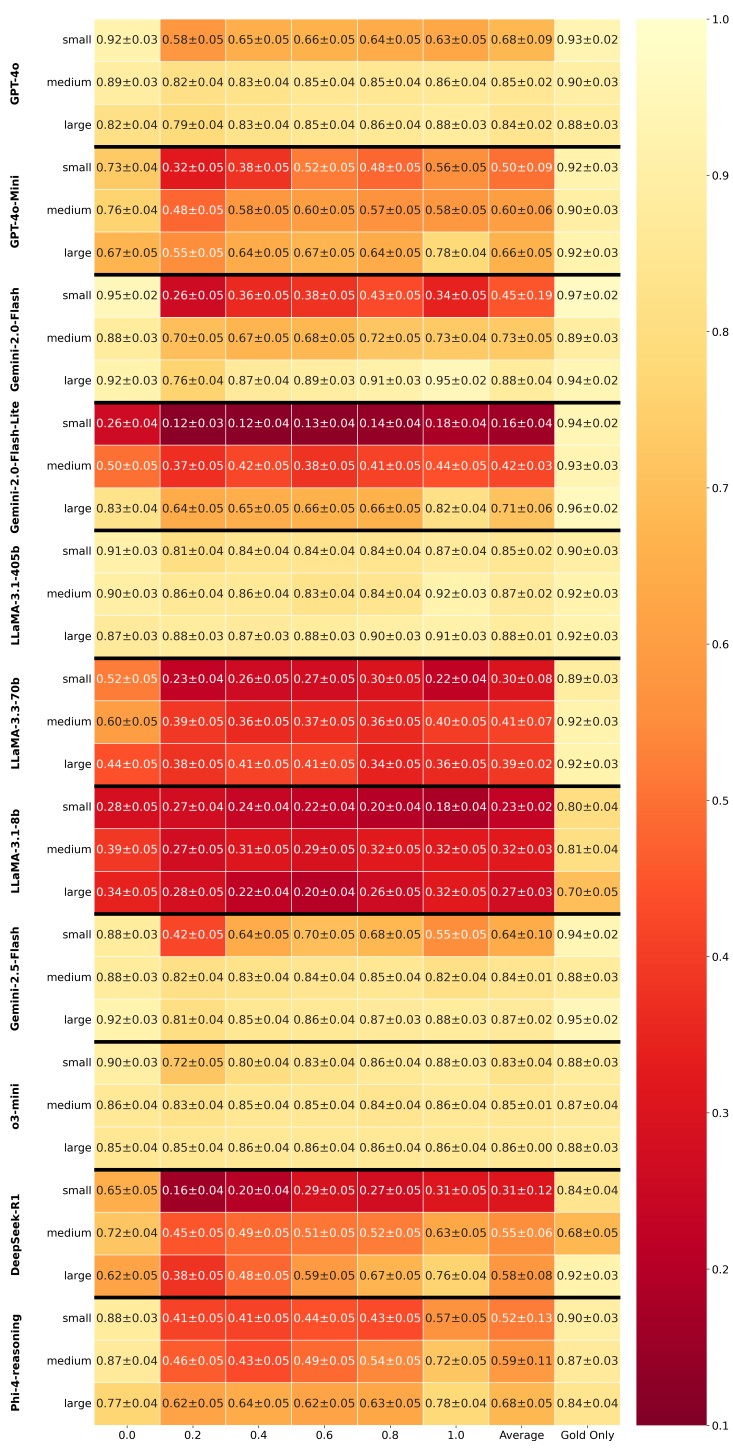

Figure 13: NuminaMath1.5 performance for each model and size of gold for varying positions in the context window (0.0, 0.2, 0.4, 0.6, 0.8, 1.0), the average across all positions, and baseline performance when seeing gold only. Higher scores (light yellow) are more desirable than low scores (dark red), 90% CI are reported.

## B.5 PERFORMANCE BY POSITION

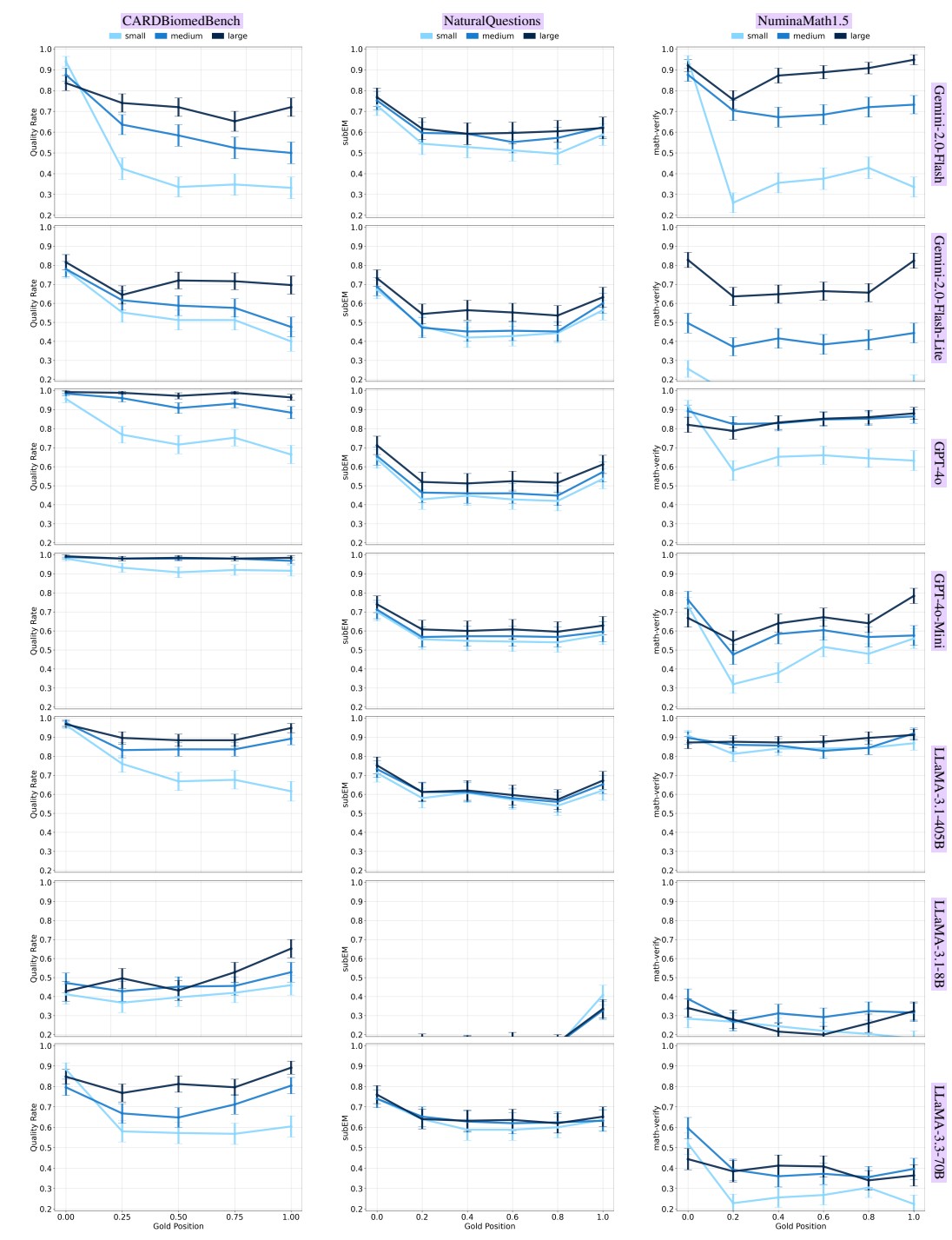

Figure 14: Model performance by gold context position (early to late in input), higher is better and error bars are 90% CIs. Each row is a model, columns are benchmarks. **Smaller gold contexts exhibit sharper performance degradation with later placement, especially in specialized domains (CBB, NM).** Larger contexts mitigate this sensitivity, highlighting the stabilizing effect of richer input. All non-reasoning models, including the ones in Figure 4, are here for comparison.

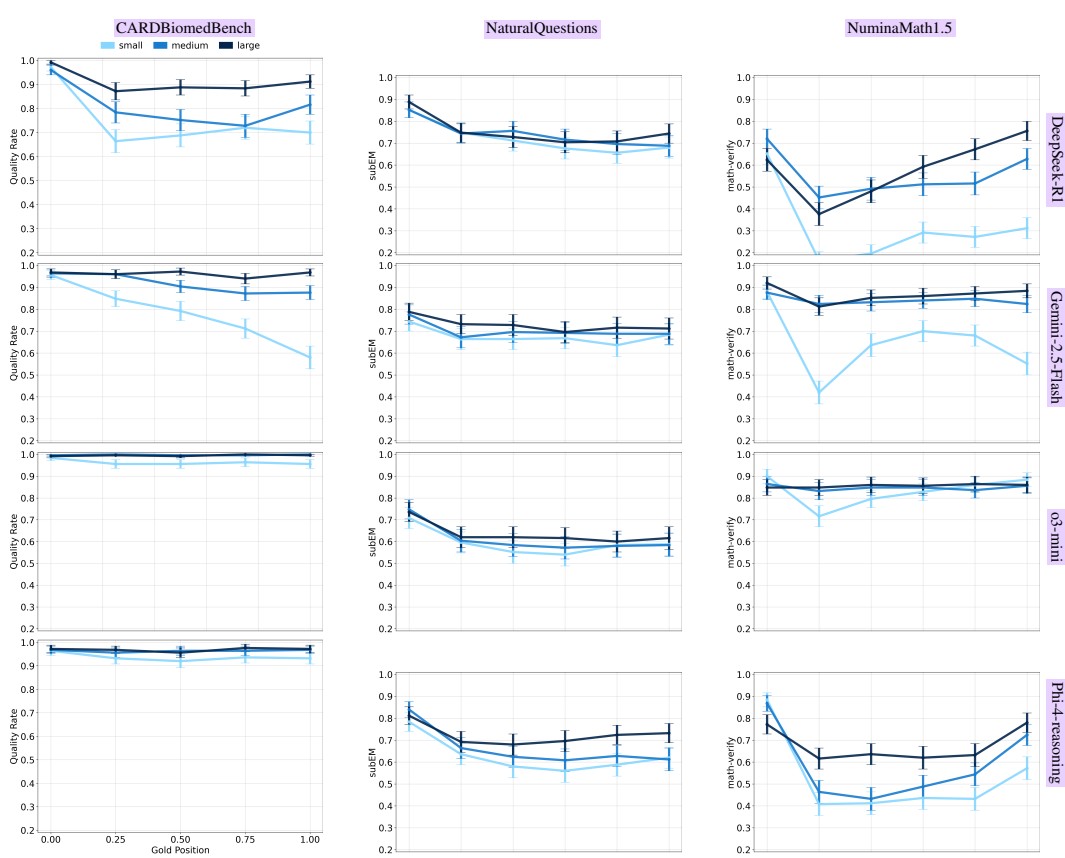

Figure 15: Reasoning model performance by gold context position (early to late in input), higher is better and error bars are 90% CIs. Each row is a model, columns are benchmarks. **Smaller gold contexts exhibit sharper performance degradation with later placement, especially in specialized domains (CBB, NM).** Larger contexts mitigate this sensitivity, highlighting the stabilizing effect of richer input.

## B.6 POSITIONAL SENSITIVITY

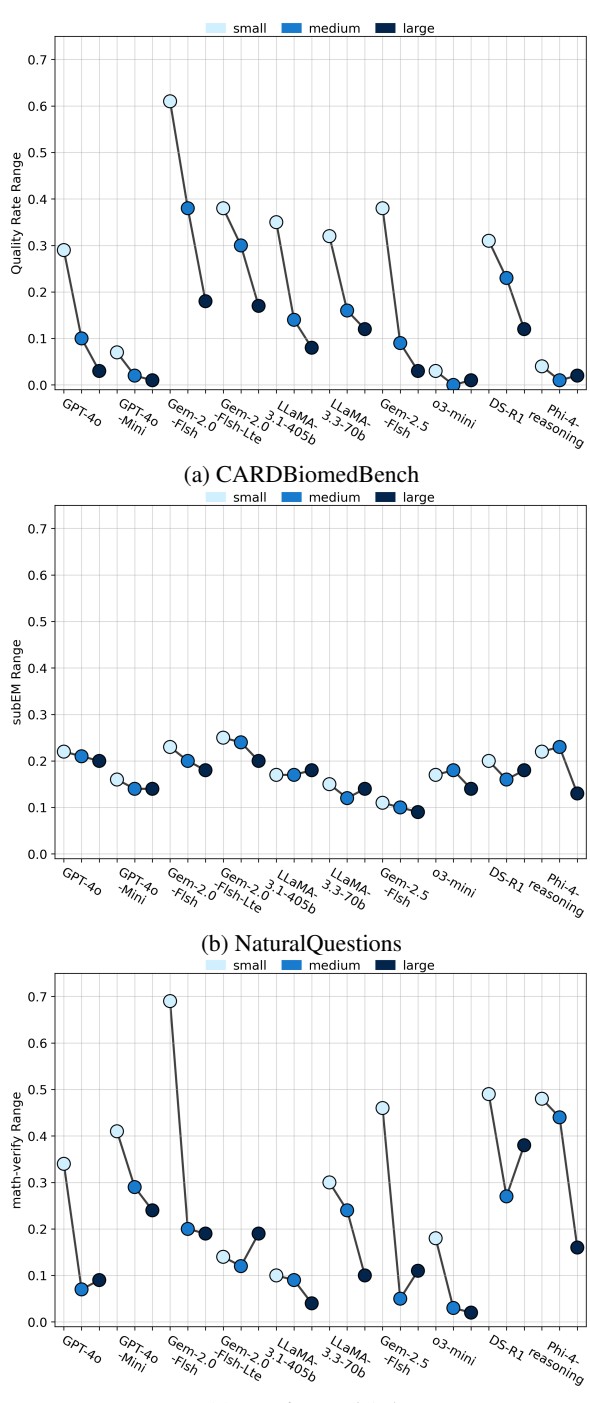

(a) CARDBiomedBench

(b) NaturalQuestions

(c) NuminaMath1.5

Figure 16: Positional sensitivity by benchmark. For each model and gold context size, we compute the range (Equation 1) of performance across positions. **Smaller gold contexts exhibit much higher sensitivity (larger ranges), especially in domain-specific tasks (CBB, NM). Larger gold contexts yield more stable performance across positions.**

## C  CONFOUNDER ANALYSIS

We provide details, formulas, distributions, and performance across benchmarks when considering the potential confounding variables.

### C.1  MEASURING GOLD CONTEXT RATIOS AND ANSWER OVERLAP

We define several metrics to quantify the repetition of the answer across gold passages, as well as the gold-to-distractor ratio.

**Gold-to-Distractor Ratio.** To measure the ratio of gold to distractor tokens, we define $T(x)$ as the tokens of passage $x$ and $|T(x)|$ is the total number of tokens in $x$:

$$\text{Gold-to-Distractor Ratio}(g, D) = \frac{|T(g)|}{\sum_{d \in D} |T(d)|} \quad (5)$$

**Exact Mentions.** We count exact string occurrences of the answer in the context, case-insensitive and word-bounded:

$$\text{ExactMentions}(a, c) = \sum_{a_i \in \mathcal{A}} \#\{\text{occurrences of } a_i \text{ in } c\} \quad (6)$$

where $\mathcal{A}$ is the set of provided answer strings and $c$ is the context.

**Answer Token Hits.** At the token level, we measure how many context tokens match any token from the answer:

$$\text{AnsTokHits}(a, c) = \sum_{t \in T(c)} \mathbf{1}[t \in T(a)] \quad (7)$$

where $T(x)$ is the tokenized version of $x$. This counts duplicates, i.e., repeated matches.

**Answer Token Repetition.**

To normalize for answer length, we define repetition as raw answer-token hits per unique answer token:

$$\text{AnsTokRepetition}(a, c) = \frac{\text{AnsTokHits}(a, c)}{|T(a)|^*} \quad (8)$$

where $|T(a)|^*$ is the number of unique tokens in the answer. This measures a normalized degree of repetition relative to the answer's own size.

Exact mentions are rare (sparse), making them an unreliable measure, thus we opt for repetition rate to capture meaningful growth in answer-token recurrence.

## C.2 CONFOUNDER DISTRIBUTIONS

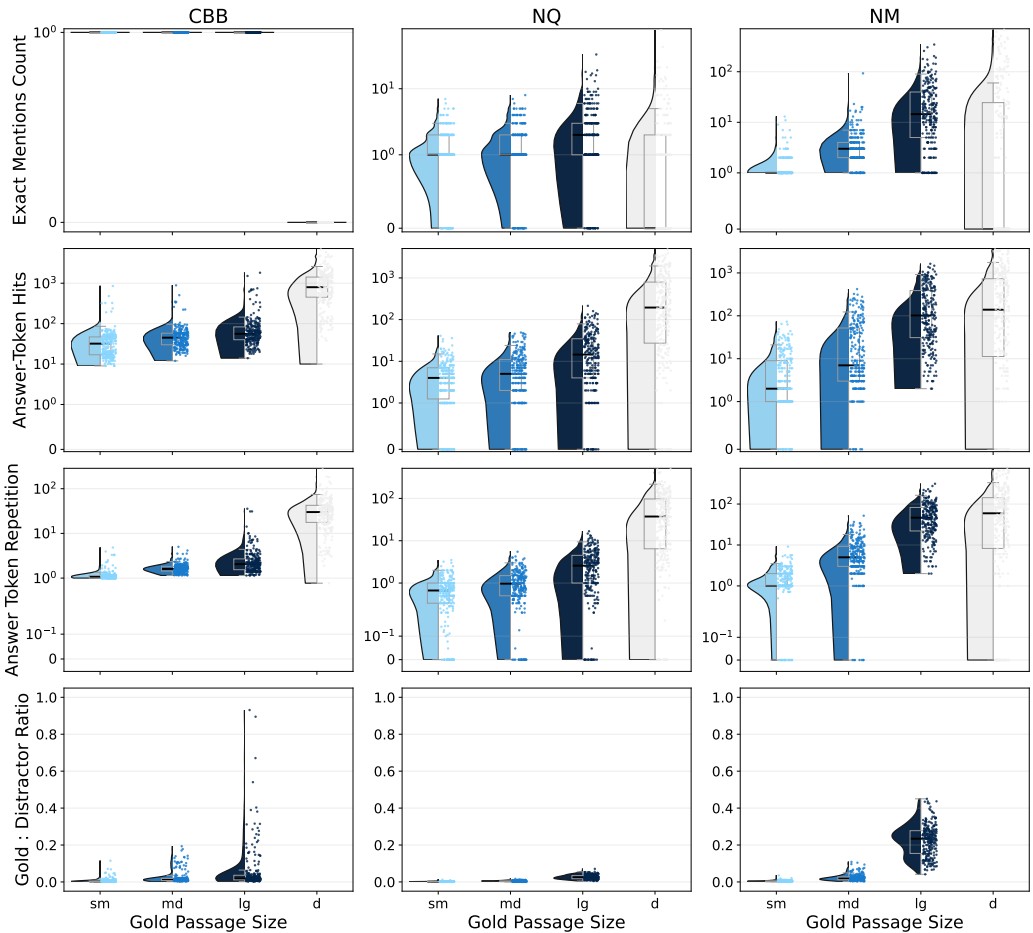

Figure 17: Raincloud plots across all benchmarks of Exact Mentions Counts, Answer-Token Hits, Redundancy, and Gold : Distractor Ratio across all sizes of gold and distractor documents for reference.

## C.3 CONFOUNDER BINNED RESULTS

**Binning Procedure.** To analyze how accuracy varies with a given feature, we partition the feature into binary bins based on the mean value across all examples (small, medium, and large).

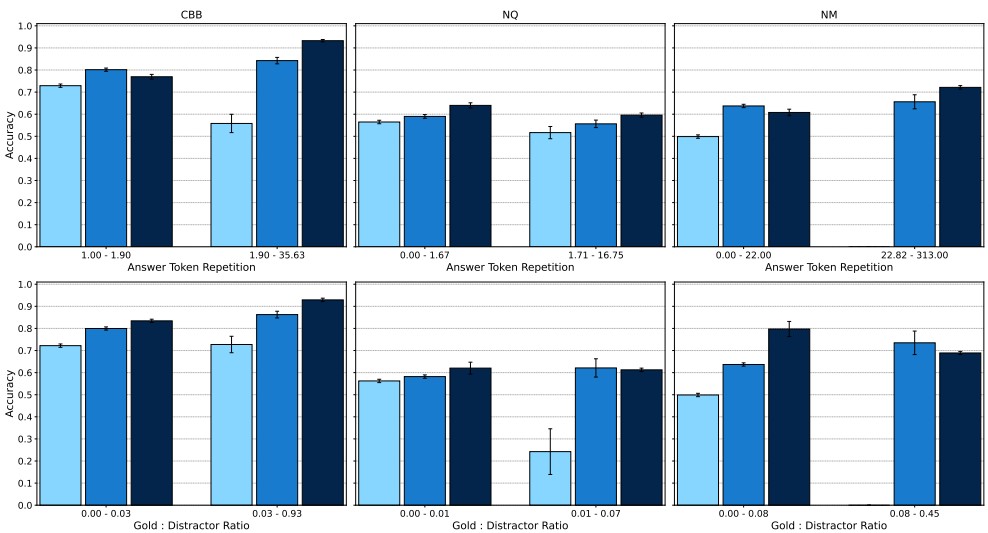

Figure 18: Performance (across all models) when bucketing tasks into binary bins per confounder: below and above the mean value. Smaller golds typically yields lower accuracy compared to larger golds. Error bars are 95% confidence intervals, some are larger due to small sample size in that bin.

## C.4 CONFOUNDER LOGISTIC REGRESSION ANALYSIS

To examine the statistical strength of each variable–gold size, answer token repetition, gold-to-distractor ratio, and position–we fit a multivariate logistic regression to predict correctness across all tasks and models for each benchmark. While the earlier analyses attempt to isolate each factor independently, these variables often co-vary in realistic long-context settings. The regression therefore provides a unified framework to estimate the partial contribution of each factor and determine whether gold context size remains an independent driver of performance.

### C.4.1 SETUP

**Motivation.**

Smaller gold contexts naturally coincide with several conditions that make retrieval harder: they tend to repeat answer tokens less often, occupy a smaller share of the input window, and are more sensitive to position. These correlations can obscure the true source of the performance gap. By modeling correctness jointly over all variables, we separate genuine size effects from those attributable to repetition, ratio, or positional bias.

**Model Specification.**

For each benchmark (CBB, NQ, NM), we regress the binary correctness outcome on four categorical predictors:

$$\Pr\big(\text{correct}\big) = \sigma(\beta_0 + \beta_{\text{size}} X_{\text{size}} + \beta_{\text{rep}} X_{\text{rep}} + \beta_{\text{ratio}} X_{\text{ratio}} + \beta_{\text{pos}} X_{\text{pos}}) \tag{9}$$

All predictors are encoded as categorical variables to keep coefficients directly interpretable–particularly for the confounders, where 'high vs. low' bins provide a clear contrast between conditions.

**Variables.** The regression includes four categorical predictors, each represented by a set of indicator variables $X_{\text{size}}$, $X_{\text{rep}}$, $X_{\text{ratio}}$, and $X_{\text{pos}}$ in the model above. Both confounder variables (repetition and ratio) are binarized into *high* vs. *low* using within-benchmark means to keep coefficients directly interpretable.

1. **Gold Size** ($X_{\text{size}}$). A categorical factor with *small* gold contexts as the reference level; the indicator variables encode the *medium* and *large* gold conditions. The coefficient $\beta_{\text{size}}$ captures the change in log-odds of correctness when moving from a small gold context to a larger one (small $\rightarrow$ medium and medium $\rightarrow$ large).

2. **Answer Token Repetition** ($X_{\text{rep}}$). Token-level overlap between the gold context and the answer, binarized at the benchmark mean into *low* (reference) vs. *high* repetition. The coefficient $\beta_{\text{rep}}$ represents the difference in log-odds between below-mean and above-mean repetition levels.

3. **Gold-to-Distractor Ratio** ($X_{\text{ratio}}$). The proportion of gold to distractor tokens in the context window, binarized at the benchmark mean into *low* (reference) vs. *high*. The coefficient $\beta_{\text{ratio}}$ quantifies the effect of moving from a lower share of gold tokens to a higher one.

4. **Position** ($X_{\text{pos}}$). The normalized start position of the gold document, treated categorically with $0.0$ (start of window) as the reference; indicator variables encode all other positions. The coefficient $\beta_{\text{pos}}$ captures how changes in placement within the input window influence correctness relative to appearing first.

**Results.**

**Across all three benchmarks, gold context size remains a significant and independent predictor of correctness after controlling for answer token repetition, gold-to-distractor ratio, and gold document position.** Medium and large gold documents consistently increase the log-odds of producing the correct answer relative to small gold contexts.

### C.4.2 CBB Logistic Regression Output

| Predictor | Coef | Std. Err. | z | P>\|z\| | 95% CI |
|---|---|---|---|---|---|
| Intercept | 1.6459 | 0.038 | 43.284 | 0.00 | [1.571, 1.720] |
| Gold Size | | | | | |
| md vs. sm | 0.3933 | 0.029 | 13.340 | 0.00 | [0.336, 0.451] |
| lg vs. sm | 0.5354 | 0.036 | 15.034 | 0.00 | [0.466, 0.605] |
| Ans Tok Rep | | | | | |
| high vs. low | 0.5924 | 0.041 | 14.309 | 0.00 | [0.511, 0.674] |
| Gold-to-Dis Ratio | | | | | |
| high vs. low | 0.3699 | 0.047 | 7.936 | 0.00 | [0.279, 0.461] |
| Position | | | | | |
| 0.25 vs. 0.0 | -0.8087 | 0.045 | -18.089 | 0.00 | [-0.896, -0.721] |
| 0.50 vs. 0.0 | -0.9001 | 0.044 | -20.305 | 0.00 | [-0.987, -0.813] |
| 0.75 vs. 0.0 | -0.8930 | 0.044 | -20.132 | 0.00 | [-0.980, -0.806] |
| 1.00 vs. 0.0 | -0.8701 | 0.044 | -19.576 | 0.00 | [-0.957, -0.783] |

Table 2: Logistic Regression Results (CBB)

Gold context size shows a clear independent effect on CBB: medium and large golds significantly increase correctness relative to small ones, even after adjusting for all confounders. Answer-token repetition and gold-to-distractor ratio also provide meaningful boosts, while position has the strongest negative impact. Performance drops sharply when the gold appears anywhere other than the start of the window. **CBB performance improves reliably with larger gold contexts, offering gains that cannot be explained by repetition, ratio, or position alone.**

### C.4.3 NQ Logistic Regression Output

| Predictor | Coef | Std. Err. | z | P>\|z\| | 95% CI |
|---|---|---|---|---|---|
| Intercept | 0.7280 | 0.027 | 26.829 | 0.00 | [0.675, 0.781] |
| Gold Size | | | | | |
| md vs. sm | 0.1129 | 0.023 | 4.992 | 0.00 | [0.069, 0.157] |
| lg vs. sm | 0.2916 | 0.051 | 5.682 | 0.00 | [0.191, 0.392] |
| Ans Tok Rep | | | | | |
| high vs. low | -0.1737 | 0.023 | -7.395 | 0.00 | [-0.220, -0.128] |
| Gold-to-Dis Ratio | | | | | |
| high vs. low | 0.0206 | 0.050 | 0.414 | 0.679 | [-0.077, 0.118] |
| Position | | | | | |
| 0.2 vs. 0.0 | -0.5486 | 0.033 | -16.795 | 0.00 | [-0.613, -0.485] |
| 0.4 vs. 0.0 | -0.5920 | 0.033 | -18.148 | 0.00 | [-0.656, -0.528] |
| 0.6 vs. 0.0 | -0.6279 | 0.033 | -19.264 | 0.00 | [-0.692, -0.564] |
| 0.8 vs. 0.0 | -0.6397 | 0.033 | -19.630 | 0.00 | [-0.704, -0.576] |
| 1.0 vs. 0.0 | -0.3815 | 0.033 | -11.601 | 0.00 | [-0.446, -0.317] |

Table 3: Logistic Regression Results (NQ)

Gold context size remains a significant predictor on NQ, though its effect is smaller than in the other benchmarks. Position exerts a substantial influence, with performance declining steadily as the gold moves deeper into the window. High answer-token repetition has a *negative effect on NQ*, and gold-to-distractor ratio shows no meaningful impact. **NQ is less sensitive to gold size and confounders than the domain-specific benchmarks, but larger gold contexts provide a consistent advantage.**

### C.4.4 NM LOGISTIC REGRESSION OUTPUT

| Predictor | Coef | Std. Err. | z | P>|z| | 95% CI |
|---|---|---|---|---|---|
| Intercept | 0.5660 | 0.028 | 20.348 | 0.00 | [0.511, 0.621] |
| Gold Size | | | | | |
| md vs. sm | 0.5675 | 0.023 | 24.926 | 0.00 | [0.523, 0.612] |
| lg vs. sm | 0.9641 | 0.080 | 12.005 | 0.00 | [0.807, 1.122] |
| Ans Tok Rep | | | | | |
| high vs. low | 0.4998 | 0.036 | 14.046 | 0.00 | [0.430, 0.569] |
| Gold-to-Dis Ratio | | | | | |
| high vs. low | -0.5138 | 0.083 | -6.220 | 0.00 | [-0.676, -0.352] |
| Position | | | | | |
| 0.2 vs. 0.0 | -0.8926 | 0.034 | -26.408 | 0.00 | [-0.959, -0.826] |
| 0.4 vs. 0.0 | -0.7459 | 0.034 | -22.000 | 0.00 | [-0.812, -0.679] |
| 0.6 vs. 0.0 | -0.6666 | 0.034 | -19.611 | 0.00 | [-0.733, -0.600] |
| 0.8 vs. 0.0 | -0.6333 | 0.034 | -18.610 | 0.00 | [-0.700, -0.567] |
| 1.0 vs. 0.0 | -0.4755 | 0.034 | -13.867 | 0.00 | [-0.543, -0.408] |

Table 4: Logistic Regression Results (NM)

NM shows the strongest size effect: medium and especially large gold contexts yield substantial gains over small ones, even after accounting for confounders. High answer-token repetition improves performance, but gold-to-distractor ratio has a negative coefficient, indicating that ratio alone does not explain the size advantage. Position is highly impactful, with penalties for placement beyond the beginning of the window. **NM is the most position-sensitive and gold-size-dependent benchmark, underscoring that larger gold contexts are crucial for stabilizing NIAH performance.**

## D  LLM DECLARATION

LLMs were used to assist in editing and revising some of the language used throughout the manuscript. Additionally, LLMs were used to edit code to create some of the figures that appear in the manuscript. The authors take full responsibility for the work in its entirety.

