# OpenReview forum: "Hidden in the Haystack: Smaller Needles are More Difficult for LLMs to Find"
_ICLR.cc/2026/Conference — Submitted to ICLR 2026_

### Official Review · Reviewer_o2Qb · 2025-10-29

**Soundness:** 3
**Presentation:** 3
**Contribution:** 2
**Rating:** 4
**Confidence:** 4

**Summary:**

This paper studies moderately long-context QA by isolating gold context size - how long the actually relevant passage is inside a long distractor-heavy prompt. The authors build controlled “haystack” prompts from biomedical QA, NaturalQuestions, and math problems, varying (i) gold passage length (small / medium / large, where each longer version strictly contains the shorter) and (ii) where in the full context that gold passage appears. They then evaluate 11 strong models (GPT-4o, Gemini-2.0-Flash, LLaMA-3.1-405B, DeepSeek-R1, etc.) across 150K+ trials.

Main result is that models are dramatically worse when the gold passage is short, especially if it appears late in the context. Longer gold passages both raise accuracy and make it less sensitive to position. In gold-only settings (no distractors), models are near-perfect, so the difficulty isn’t the task itself — it’s finding and using the right span in a sea of noise. The authors argue this matters for retrieval-augmented / agentic systems that merge evidence from multiple tools.

**Strengths:**

1. The experimental setup is clean: same question, same distractors, but systematically varying gold length and position.  The baselines (closed-book, gold-only, distractor-only) are helpful controls.

2. Across domains and models, the pattern is stable. Longer gold ⇒ higher accuracy, shorter gold ⇒ positional fragility. For example, in biomedical QA, accuracy jumps 20–30 points going from “small” to “large” gold for leading models, and late-placed short gold is often ignored almost completely. This is consistent across biomed, web QA, and math.

3. The takeaway is operational. If you pass a model only a tiny “critical snippet” from one tool alongside lots of irrelevant but plausible-looking text from other tools, it may miss it. That’s highly actionable for system designers.

**Weaknesses:**

1. The headline conclusion that the more/denser relevant information you show the model, the more likely it is to answer correctly; tiny late snippets get ignored is intuitively what most people would predict. The paper proves this rigorously and at scale, which is valuable, but the scientific novelty feels modest.

2. The authors hypothesize that longer gold passages “attract attention” and reduce positional brittleness, but don’t dig into internal attention patterns or multi-gold scenarios (e.g. several short relevant snippets spread across the prompt). The current analysis is observational rather than explanatory.

3. The study inserts a single gold passage at controlled positions among distractors, but real deployments use RAG pipelines where retrieval ranking determines position and where multiple semi-relevant chunks, reranking, chunk sizes, and length-normalized scoring all interact. An end-to-end RAG setting (with retriever errors, ranking-induced positions, and multi-gold evidence) would test whether the “short gold gets ignored” effect persists under realistic ranking—and whether strategies like padding/expanding short gold, reranking by gold-length proxies, or top-k fusion mitigate it. This missing evaluation limits external validity.

**Questions:**

1. Does distributing multiple short gold snippets across early/mid/late approximate the robustness of one long gold block, or is contiguity essential?

2. Can you provide any model-internal evidence (e.g. attention focus) for the “longer gold is harder to ignore” story?

---

> ### Author Response · Authors · 2025-11-17
>
> We thank the reviewer for this thoughtful and balanced evaluation. We especially appreciate the positive remarks highlighting that our “experimental setup is clean,” the consistency of results across domains and models, and the “operational” relevance of the findings. We are glad the reviewer recognizes that our framework enables a controlled examination of gold context size in a way prior work has not explored.
>
> As a reminder, here are the main contributions of our work: Prior work has analyzed positional bias, distractor count, and attention dynamics, but—as the reviewer notes—does not propose that the relationship between **gold context size** and long-context performance is an important factor. Our setup holds the *task, answer, question, and distractors constant* while varying only the size of the *same* gold evidence, allowing us to expose a robust finding: **smaller gold documents are disproportionately missed and exhibit extreme sensitivity to position**, and this persists even after controlling for five potential confounders. This reveals an important fragility in long-context aggregation—one that appears across biomedical reasoning, math, and general QA, and across eleven modern LLMs.
>
> Below, we respond to the concerns raised.
>
> > W1) the scientific novelty feels modest.
> >
>
> **TLDR: The finding is a more nuanced than “more info helps”—short but sufficient evidence becomes fragile when mixed with longer outputs, a real and previously unexamined failure mode. This holds even after controlling for repetition, distractor volume, and ratio.**
>
> **We un**derstand this concern and appreciate the chance to clarify why the contribution is more substantial than “more relevant information helps.”
>
> **Why this matters in practice:**  In real agentic deployments, retrieval and tool outputs are heterogeneous: some components return short snippets while others emit long summaries or multi-paragraph rationales. Critically, **systems often cannot know in advance which piece of evidence will be the “gold”**, yet all pieces are concatenated together. Our findings show that under such conditions, (1) short-but-critical evidence can be *systematically overshadowed* by longer, semantically rich outputs, and (2) this failure emerges *even when the short snippet alone is fully sufficient to answer the question*. This has direct operational consequences for multi-tool, multi-agent, and retrieval-augmented architectures, where evidence mixing is unavoidable and length heterogeneity is the norm—not a corner case.
>
> **Why “larger is better” is *not* obvious under our controls:** We intentionally designed a strict experimental setting to avoid trivial explanations:
>
> - **All gold versions (small/medium/large) are individually sufficient** to solve the task in isolation (gold-only baselines).
> - **Larger golds do *not* simply repeat the answer**, as controlled in §4.2.
>
> Once these factors are fixed, the naïve expectation “larger = higher quality = easier” is not guaranteed. In fact, if all gold versions convey the same minimal answer, one might reasonably expect the smallest version (the most concise signal) to be easier to detect, not harder.
>
> Thus, the empirical observation is not a trivial restatement of “longer is clearer.”
>
> **Why the size–position interaction is novel:** Our ablations show that even after controlling for repetition, distractor volume, gold-to-distractor ratio, and domain, **short gold suffers an extreme positional brittleness**:
>
> - When small gold appears late, models often ignore it almost completely.
> - Larger gold dramatically reduces positional sensitivity.
> - This holds across biomedical reasoning, math, and open-domain QA, and across 11 strong LLMs.
>
> Prior long-context evaluations typically assume equal-length passages or vary distractor count; none examine how *the relative size of the same relevant evidence* fundamentally changes positional robustness.
>
> *reply continued to the next comment*

---

> ### Author Response · Authors · 2025-11-17
>
> > W2) dig into internal attention patterns
> Q2) Can you provide any model-internal evidence (e.g. attention focus) for the “longer gold is harder to ignore” story?
> >
>
> **TLDR: We can probe representations or inspect attention maps, but both are correlational; we welcome any concrete suggestions for deeper analysis.**
>
> We appreciate this suggestion — the spirit of this comment resonates with us! At a high level, we’d love to have a better understanding of model internals and how they lead to the observed outcome here, for example, by isolating “failure to attend” from “failure to integrate.” Unfortunately, the underlying mechanisms at work are difficult to disentangle in LLMs — not only in our setting but for *any* long-context or reasoning task.
>
> That said, we have considered two possible analyses:
>
> - **Probing intermediate representations:** Prior work (e.g., [1]) shows that models often encode the correct answer internally even when they fail to express it. One could apply a similar probing to our setup to test whether small-gold failures reflect “known but unused” information and track the probe scores as the gold length increases. While potentially informative, such probes are correlative and highly distribution-dependent, making it hard to draw conclusive lessons.
> - **Visualizing/analyzing attention patterns:** One could also compare attention distributions (e.g., final-layer attention mass on gold vs. distractors) between small and large gold conditions. This would indicate whether small gold contexts receive systematically less attention. However, attention inspection is also observational, and changes in attention mass do not necessarily imply causal influence on model decisions (various works have discussed this “attention vs explanation” question [2]).
>
> We are happy to run either analysis if the reviewer believes it would add value. However, we expect both would remain correlational and would not definitively separate "attention failure" from "aggregation failure"—a fundamental challenge beyond the scope of this work. If the reviewer has suggestions for analyses that could better probe this distinction, we welcome them and would be glad to incorporate additional experiments.
>
> [1]  Insights into LLM Long-Context Failures: When Transformers Know but Don't Tell. 2024 https://arxiv.org/abs/2406.14673
>
> [2] Is Attention Explanation? An Introduction to the Debate, 2022 https://aclanthology.org/2022.acl-long.269.pdf
>
> *reply continued to the next comment*

---

> > ### Author Response · Authors · 2025-11-18
> >
> > > W3) Real end-to-end retriever-based systems have more components to determine position (reranking, scoring, etc).
> > >
> >
> > **TLDR: Our finding applies to any in multi-agent (DeepResearch-esque) system that aggregate evidence  (some short and some long), making this failure mode directly relevant.**
> >
> > We appreciate this insightful comment. We view the scope of our controlled setup as **broader than classical RAG**. Many agentic systems today solve problems by **mixing evidence from heterogeneous sub-agents (tools, search, tables, etc.) without knowing in advance which piece will be gold and not necessarily controlling for length**.
> >
> > Our finding—that **short-but-critical evidence is disproportionately fragile, especially when placed late**—directly applies to these agentic pipelines. In fact, such systems often amplify exactly the failure mode we surface: short, high-signal evidence from a specialized sub-agent gets overshadowed by long reasoning traces or summaries produced by others. This failure mode is broader than RAG systems and is fundamental to any multi-component system that merges variable-length evidence.
> >
> > Because there is enormous diversity in how these systems are built, our goal—consistent with prior work—is to study the underlying behaviors **in isolation**, independent of any particular architecture, so that the findings are general. Constructing a full end-to-end system with multiple sub-agents is a substantial engineering effort and beyond the scope of this paper, but we fully agree that it is a valuable direction for future work.
> >
> > > Q1) Does distributing multiple short gold snippets across early/mid/late approximate the robustness of one long gold block, or is contiguity essential?
> > >
> >
> > **TLDR: Splitting gold harms coherence and changes the task; we are open to exploring a framing that preserves meaning without introducing confounders.**
> >
> > Thank you for the thoughtful suggestion. We considered this diagnostic experiment during our original submission and ultimately decided against it, as fragmenting the gold document into multiple dispersed parts sacrifices its *contiguity*—a key property of any natural language document. When we split a coherent gold passage into scattered snippets, we risk altering its meaning, coherence, and retrievability, effectively turning the task into a different one. This makes it difficult to isolate the effect of mention quantity from confounding factors like incoherence.
> >
> > That said, we agree there may be a way to implement this diagnostic without introducing confounding factors. If you have a specific framing in mind that better aligns with our core hypothesis, we’d be glad to explore it experimentally.
> >
> > ---
> >
> > **We look forward to addressing any remaining questions during the rebuttal and hope these updates warrant a higher evaluation.**

---

> > > ### Comment · Reviewer_o2Qb · 2025-11-27
> > > **Results are solid but contribution is not sufficient**
> > >
> > > Thank you for the thorough and thoughtful rebuttal. Your motivation to understand mechanisms behind long-context failures came through well in your response.
> > >
> > > My overall assessment remains that - while the results are technically solid - the contribution in its current form feels somewhat limited for a conference of this level. I believe the paper would benefit from two additions:
> > >
> > > 1. End-to-end RAG baselines.
> > >
> > >    Because your core claim has immediate implications for retrieval-augmented systems, evaluating whether the same “short gold is fragile” effect persists under realistic retriever–ranker pipelines would significantly improve external validity.
> > >
> > > 2. Mechanistic analysis.
> > >
> > >    Your results raise genuine questions about why short gold is overlooked. Adding even lightweight mechanistic analysis would make the story more explanatory rather than purely observational. It does not need to be definitive; even partial evidence would sharpen the contribution.
> > >
> > > If these extensions are incorporated, I believe the paper’s overall impact would rise meaningfully.

---

> > > > ### Author Response · Authors · 2025-12-01
> > > >
> > > > Thank you for the follow-up and for engaging deeply with our rebuttal. We appreciate the acknowledgment of the technical solidity of the results and of our effort to articulate the underlying motivation.
> > > >
> > > > We also appreciate the reviewer’s suggestions for future extensions. However, we respectfully disagree that the contribution, in its current form, is limited without these additions.
> > > >
> > > > We clarify our position below.
> > > >
> > > > **1. On the request for end-to-end RAG baselines.**
> > > >
> > > > Our evaluation already contains an end-to-end RAG evaluation. Specifically, for the **NQ benchmark**, distractor documents were **retrieved** from an index of Wikipedia articles using a top-tier embedding-based retriever `gte-large-en-v1.5` [4]. In this setting, the model must identify the correct answer from among a collection of retrieved documents, which is the core of a RAG pipeline.
> > > >
> > > > At the same time, we reiterate that our claims are **not limited to classical RAG**. The failure mode we identify arises whenever heterogeneous evidence (short snippets, long rationales, multi-step traces) is concatenated, which is common in modern **multi-agent and multi-tool systems** (e.g., DeepResearch-style planners, multi-agent biomedical assistants, reasoning toolchains). In these systems, the gold evidence is *not known a priori*, and the mixing of variable-length outputs is unavoidable. Our setup was intentionally designed to reflect these real deployments.
> > > >
> > > > Since you also hinted at having a “realistic” setup, **all 3 of our benchmarks are strongly grounded in real use cases**. In each of the benchmarks, the questions are sourced from specific real settings:
> > > >
> > > > - **CARDBiomedBench (CBB):** Questions are authored by *biomedical research experts* and reflect genuine clinical and biomedical research information needs [1].
> > > > - **Natural Questions (NQ):** Questions come from *real Google Search queries* typed by end users [2].
> > > > - **NuminaMath 1.5 (NM):** Problems are competition-style math questions requiring multi-step reasoning [3].
> > > >
> > > > Just as importantly, we ensured that **distractors** simulate realistic retrieval noise—factual, topically relevant, and plausibly confusable:
> > > >
> > > > - **CBB:** Distractors are retrieved from a deployed multi-agent biomedical LLM system. The agents retrieve from domain-specific sources such as HGMD, PubTator3.0, NCBI gene/variant pages, and Google Search, scoped to NIH domains.
> > > > - **NQ:** Distractors **retrieved** from an index of Wikipedia articles using a top-tier embedding-based retriever.
> > > > - **NM:** Distractors were randomly sampled reasoning traces to different questions, simulating a noisy but topically relevant set of distracting documents.
> > > >
> > > > Collectively, these choices give us a robust long-context evaluation environment that is *already* RAG-like — and, importantly, generalizes to a broader class of agentic pipelines that mix heterogeneous evidence.
> > > >
> > > > [1]  https://pubmed.ncbi.nlm.nih.gov/39868292/
> > > >
> > > > [2] https://aclanthology.org/Q19-1026/
> > > >
> > > > [3] https://huggingface.co/datasets/AI-MO/NuminaMath-1.5
> > > >
> > > > [4] https://huggingface.co/Alibaba-NLP/gte-large-en-v1.5
> > > >
> > > > ---
> > > >
> > > > **2. On the request for a mechanistic analysis.**
> > > >
> > > > We appreciate the desire for more mechanistic insight. While we agree with the aspiration to provide a *causal* explanation of model behavior, *this goal is itself an active area of research.* It lies beyond the scope of our stated goal. Existing methods (attention maps, value-flow tracing, probing, activation patching) are largely correlational rather than causal.
> > > >
> > > > In our original rebuttal, we asked whether the reviewer had a specific mechanistic experiment in mind. The response did not specify any concrete or actionable proposal. We would still welcome clarification: **what type of mechanistic analysis, ideally with a causal interpretation, would you consider sufficient to accept the empirical finding?**
> > > >
> > > > In our view, developing causal mechanistic accounts is a natural and valuable direction, *building on top of* our empirical findings, and we have clarified in the Discussion that we see this as important future work rather than a requirement for the present contribution
> > > >
> > > > ---
> > > >
> > > > In summary, we appreciate the reviewer’s suggestions and have incorporated additional clarifications to acknowledge their value as future work. At the same time, we maintain that the current submission offers a substantive and novel contribution: *it isolates a fundamental long-context fragility that has not been documented in prior work, demonstrates it at scale across domains and models, and provides actionable insights*.

---

### Official Review · Reviewer_9NSP · 2025-10-30

**Soundness:** 2
**Presentation:** 2
**Contribution:** 3
**Rating:** 6
**Confidence:** 3

**Summary:**

This paper investigates the impact of gold context length on the performance of LLMs in Needle-in-a-Haystack (NIAH) tasks. Through controlled experiments across multiple LLMs and different benchmarks, the paper draws a clear conclusion, i.e., LLM performance in NIAH tasks degrades significantly when the gold context is shorter. The study demonstrates that even after controlling for confounding variables such as gold context position, answer repetition rate, and distractor volume, gold context length remains a significant independent predictor of success. The research also finds that smaller gold contexts amplify the model's positional sensitivity.

**Strengths:**

1. This paper proposes an interesting and intuitive factor affecting NIAH performance, i.e., the gold context length. The study of this factor breaks down the reliability of the notion that "longer inputs always lead to performance degradation" in practical scenarios.
2. The paper conducts extensive experiments to demonstrate that gold context length reduces LLM performance in NIAH tasks and that this performance is more sensitive to position.

**Weaknesses:**

1. The most significant issue is that this paper only points out the problem without conducting a mechanistic analysis. This makes it difficult to determine whether this is a temporary issue or a fundamental deficiency in LLMs. Furthermore, the paper does not test on the latest models known for excellent long-context performance, such as Gemini 2.5 Pro and Claude. Notably, Figure 3 shows that o3-mini is less affected by changes in context length, which heightens my concern about the significance of the problem raised by the paper .
2. A minor concern: Performance on the NIAH task may not necessarily correlate with real-world LLM in-context performance. Therefore, the lack of validation of this phenomenon on realistic tasks makes it difficult to judge the problem's importance.
3. The gold contexts used in this paper differ not only in length but also in content type . This implies that the performance shown in Figure 3 could also be caused by variations in content quality
4. The x-axis labels in some figures are rotated, making it difficult to clearly locate the corresponding data, such as in Figure 3 and Figure 7.

**Questions:**

1. Can the authors demonstrate whether this problem remains significant for state-of-the-art models, such as Gemini 2.5 Pro and Claude?
2. I noticed in Figure 3 that for some models and tasks, the "medium" group performed better than the "large" group (e.g., o3-mini in (a) and o3 in (c)). This phenomenon slightly contradicts the authors' conclusions. Can the authors provide an analysis of this anomaly?
3. Why was the repetition rate metric from Eq. (1) used for the experiment decoupling answer repetition and gold context length, rather than other repetition metrics, such as the more intuitive "Exact Mentions" (as defined in Eq. (7))?

---

> ### Author Response · Authors · 2025-11-18
>
> We thank the reviewer for the thoughtful and constructive evaluation. We are grateful for the positive remarks, including: *“This paper proposes an interesting and intuitive factor affecting NIAH performance,” “conducts extensive experiments,” a*nd *“breaks down the reliability of the notion that ‘longer inputs always lead to performance degradation.’”* We appreciate the reviewer’s recognition that the work is interesting, clear, and adds insight to the field by challenging common assumptions.
>
> As a reminder, here are the main contributions of our work: prior work has analyzed positional bias, distractor count, and attention dynamics, but—as the reviewer notes—does not propose that the relationship between **gold context size** and long-context performance is an important factor. Our setup holds the *task, answer, question, and distractors constant* while varying only the size of the *same* gold evidence, allowing us to expose a robust finding: **smaller gold documents are disproportionately missed and exhibit extreme sensitivity to position**, even after controlling for five potential confounders. This reveals an important fragility in long-context aggregation—one that appears across biomedical reasoning, math, and general QA, and across eleven modern LLMs.
>
> We now address your concerns:
>
> > W1) Only points out the problem without conducting a mechanistic analysis. This makes it difficult to determine whether this is a temporary issue or a fundamental deficiency in LLMs.
> >
>
> **TLDR: Disentangling attention vs aggregation mechanistically is fundamentally hard; we welcome any concrete suggestions for deeper analysis.**
>
> **We ap**preciate this concern and agree it is important. Unfortunately, we believe the underlying mechanisms at work—*attention* and *aggregation*—are inherently difficult to disentangle in modern LLMs. The attention mechanism itself simultaneously (1) allocates focus across tokens and (2) aggregates information through value vectors; this process is repeated across many heads and layers with potentially heterogeneous behaviors. As a result, isolating “failure to attend” from “failure to integrate” is not straightforward, not only in our setting but for long-context reasoning more broadly.
>
> That said, we have considered two possible analyses:
>
> - **Probing intermediate representations:** Prior work (e.g., [1]) shows that models often encode the correct answer internally even when they fail to express it. One could apply a similar probing to our setup to test whether small-gold failures reflect “known but unused” information and track the probe scores as the gold length increases. While potentially informative, such probes are correlative and highly distribution-dependent, making it hard to draw conclusive lessons.
> - **Visualizing/analyzing attention patterns:** One could also compare attention distributions (e.g., final-layer attention mass on gold vs. distractors) between small and large gold conditions. This would indicate whether small gold contexts receive systematically less attention. However, attention inspection is also observational, and changes in attention mass do not necessarily imply causal influence on model decisions (various works have discussed this “attention vs explanation” question [2]).
>
> We are happy to run either analysis if the reviewer believes it would add value. However, we expect both would remain correlational and would not definitively separate "attention failure" from "aggregation failure"—a fundamental challenge beyond the scope of this work. If the reviewer has suggestions for analyses that could better probe this distinction, we welcome them and would be glad to incorporate additional experiments.
>
> *reply continued to the next comment*

---

> > ### Author Response · Authors · 2025-11-18
> >
> > > W2) NIAH task may not correlate with real-world LLM in-context performance. Lack of validation of this phenomenon on realistic tasks makes it difficult to judge the problem's importance.
> > >
> >
> > **TLDR:** **We validate on benchmarks from 3 distinct domains that are strongly aligned with real use cases.**
> >
> > We agree that validating on realistic tasks is important, and this is precisely why we deliberately chose benchmarks that are strongly synergistic with real-world use cases (Appendix A.1.). In each of the benchmarks, the questions are sourced from specific real settings:
> >
> > - **CARDBiomedBench (CBB):** Questions are authored by biomedical research experts, reflecting genuine clinical and biomedical-research information needs [1].
> > - **Natural Questions (NQ):** Questions come from *real Google Search queries* typed by end users [2].
> > - **NuminaMath 1.5 (NM):** Problems are competition-style math questions requiring multi-step reasoning [3].
> >
> > Just as importantly, we ensured that **distractors** simulate realistic retrieval noise—factual, topically relevant, and plausibly confusable:
> >
> > - **CBB:** Distractors are retrieved from a deployed multi-agent biomedical LLM system. The agents retrieve from domain-specific sources such as HGMD, PubTator3.0, NCBI gene/variant pages, and Google Search, scoped to NIH domains.
> > - **NQ:** Distractor documents were retrieved from an index of Wikipedia articles using a top-tier embedding-based retriever `gte-large-en-v1.5` [4].
> > - NM: Distractors were randomly sampled reasoning traces to different questions, simulating a noisy but topically relevant set of distracting documents.
> >
> > Together, these choices ensure that the phenomenon we identify is not an artifact of synthetic NIAH benchmarks but **robustly present in realistic, domain-grounded scenarios** across biomedical, general QA, and mathematical reasoning.
> >
> > [1]  https://pubmed.ncbi.nlm.nih.gov/39868292/
> >
> > [2] https://aclanthology.org/Q19-1026/
> >
> > [3] https://huggingface.co/datasets/AI-MO/NuminaMath-1.5
> >
> > [4] https://huggingface.co/Alibaba-NLP/gte-large-en-v1.5
> >
> > > W3) Gold contexts differ in both length and content type; variations can be due to quality.
> > >
> >
> > **TLDR: Gold-only baselines show all sizes are equally solvable; larger golds do not simply repeat the answer more, so quality is not driving the effect.**
> >
> > We agree that "evidence quality" is an important dimension for analysis. Longer gold documents may be higher quality—they can provide contextualization or rationalization for the answer. However, characterizing this "richness" in a formal or quantifiable way is challenging.
> >
> > In our approach, the goal was to maintain objective choices; we followed two principles:
> >
> > - **Gold-only baselines are equally strong:** We controlled for answer richness via gold-only baselines (Appendix B.2). All gold versions (small/medium/large) yield **uniformly high accuracy in isolation** when shown alone, without distractors. This demonstrates that **the smallest gold contains all the information needed to solve the task.** Therefore, differences in performance under distractors are not due to insufficient content or reasoning completeness.
> > - **Larger gold documents do not repeat the answer more:** We measure answer token repetition and group tasks into comparable repetition ranges (§4.2). Within each repetition bucket, larger gold contexts still outperform smaller ones. This shows that the advantage of larger contexts is not merely due to having "more copies" of the answer—it cannot be reduced to a trivial signal-amplification explanation.
> >
> > We believe we have controlled for richness as thoroughly as possible within the constraints of naturally occurring datasets. If the reviewer has suggestions for how to operationalize or measure quality more directly, we would welcome them.
> >
> > > W4) Figure 3 and Figure 7 hard to read.
> > >
> >
> > We appreciate the suggestion. The figures have been redesigned for improved legibility.
> >
> > *reply continued to the next comment*

---

> ### Author Response · Authors · 2025-11-18
>
> > Q3) Why use repetition rate rather than exact mentions?
> >
>
> **TLDR: Exact mentions are rare (sparse), making them an unreliable measure; repetition rate captures meaningful growth in answer-token recurrence and helps rule out trivial amplification.**
>
> We use *repetition rate* rather than *exact mentions* because our goal is to measure whether larger gold documents contain more occurrences of the answer tokens. If so, performance differences would be easier to explain: repeated tokens effectively encode the answer multiple times and increase the model’s chances of attending to it. While both metrics capture repetition, exact mentions are infrequent and show little variation across gold-context sizes.
>
> The table below shows the distribution of **`exact_mentions`** (median, IQR [Q1, Q3]) across gold context sizes and distractors:
>
> | **Benchmark** | **Small Gold** | **Medium Gold** | **Large Gold** | **Distractor** |
> | --- | --- | --- | --- | --- |
> | CBB | 1.0 [1.0, 1.0] | 1.0 [1.0, 1.0] | 1.0 [1.0, 1.0] | 0.0 [0.0, 0.0] |
> | NQ | 1.0 [1.0, 2.0] | 1.0 [1.0, 2.0] | 2.0 [1.0, 3.0] | 0.0 [0.0, 2.0] |
> | NM | 1.0 [1.0, 1.0] | 3.0 [2.0, 4.0] | 14.5 [5.0, 39.8] | 0.0 [0.0, 24.5] |
>
> **These results indicate that exact answer span repetition does not substantially increase with gold document size in most cases and therefore is not the cause of performance increase.** In CBB and NQ, median counts remain stable at 1.  Only NM shows a notable rise, driven by numerical answers that commonly recur in the documents. These distributions are plotted in Appendix C.2.
>
> The next table shows the distribution of **`repetition_rate`**—the ratio of overlapping answer tokens to answer length:
>
> | **Benchmark** | **Small Gold** | **Medium Gold** | **Large Gold** | **Distractor** |
> | --- | --- | --- | --- | --- |
> | CBB | 1.1 [1.0, 1.1] | 1.6 [1.4, 1.8] | 2.1 [1.5, 2.7] | 30.0 [17.6, 42.5] |
> | NQ | 0.7 [0.3, 1.0] | 1.0 [0.5, 1.5] | 2.6 [1.0, 4.5] | 37.3 [6.5, 96.9] |
> | NM | 1.0 [1.0, 2.0] | 5.0 [3.0, 9.3] | 46.4 [22.0, 82.6] | 142.9 [59.4, 711.0] |
>
> **These results indicate that as document length increases, answer token repetition** **becomes more common and may confound results.** This metric, therefore, serves as a meaningful way to decouple answer token repeats from gold document size.
>
> This prompted our deeper analysis and conclusions (§4.2 and Figure 5) that, **although repetition does occur, as measured by the repetition rate, our claim that small gold documents are harder to find still holds.**
>
> ---
>
> **We look forward to addressing any remaining questions during the rebuttal and hope these updates warrant a higher evaluation.**

---

### Official Review · Reviewer_hRbP · 2025-10-30

**Soundness:** 2
**Presentation:** 3
**Contribution:** 2
**Rating:** 4
**Confidence:** 4

**Summary:**

This paper investigates how the size of the relevant context (the "needle") affects an LLM's performance in a needle-in-a-haystack setting. The authors conduct a large-scale study by creating three versions of each "gold" document (small, medium, large) and embedding them at various positions within a fixed-size haystack of distractors.

The core findings are:
1. LLM performance drops significantly when the gold context is smaller.
2. This "size effect" is strongly linked to positional bias. Smaller needles are far more sensitive to their position and show an extreme primacy bias (i.e., they are only found reliably at the very beginning of the context).
3. The authors perform several analyses to argue that this effect is not an artifact of answer repetition or distractor volume.

**Strengths:**

1. The study is comprehensive. The authors conducted a sheer scale of the experiments, testing 11 modern LLMs (including strong proprietary and open-weight models) on three diverse tasks (biomedical, general QA, and math). This provides strong evidence that the findings are not a model-specific or task-specific fluke.
2. The paper's most valuable contribution is not just that "size matters," but its demonstration of the interaction between gold context size and positional bias.
3. The findings are useful for anyone building real-world retrieval-augmented systems. It directly challenges the common wisdom that retrieving the smallest, most minimal-and-relevant chunk is always the best strategy. This work provides a clear, data-driven reason why that approach might fail.

**Weaknesses:**

My main concerns are with the experimental design, which seems to entangle the core variable of interest ("absolute size") with other, more powerful confounding variables.
1. Entangled variables (size vs. ratio): The study fixes the distractor (haystack) size. This means gold_context_size and gold_to_distractor_ratio are perfectly correlated. A "small needle" is always a "low signal-to-noise ratio," and a "large needle" is always a "high ratio." The post-hoc analysis in Sec 4.3 does not adequately de-couple these variables. The results could just as easily support the less novel conclusion that "lower signal-to-noise ratios are harder to find."
2. Conflated variables (size vs. quality): The gold contexts (see Fig 8) are not just different in size; they differ in quality. The "large" gold context contains the "complete reasoning process," while the "small" one has only the "minimal span." The experiment is thus conflating size with information richness. It's unsurprising that models perform better when given a more comprehensive, "better" answer, which just also happens to be larger.

**Questions:**

1. Given the entanglement of absolute size and gold_to_distractor_ratio, why are the authors confident that absolute size is the determining factor, rather than the (in my view, more likely) signal-to-noise ratio? Maybe a small-scale experiment can be conducted to truly isolate the variables.
2. Could the authors clarify the mechanism of failure? When a model fails on a "small needle," is it an attention failure (the model's attention mechanism literally does not "see" the relevant tokens) or an aggregation failure (the model sees the small needle but judges it as less credible or important than a larger, more "authoritative-looking" distractor document)?
3. Related to Weakness #2: Can you defend the choice to make the larger contexts also contain more comprehensive supporting information? How can you be sure that the performance gain doesn't simply come from this higher quality of evidence, rather than the size of the context it's embedded in?

---

> ### Author Response · Authors · 2025-11-17
>
> We thank the reviewer for the thoughtful and detailed evaluation. We are grateful for the positive remarks, including: *“the study is comprehensive,”* *“the paper's most valuable contribution is the interaction between gold context size and positional bias,”* and *“the findings challenge common wisdom for retrieval-augmented systems.”* We appreciate the recognition that the scale, scope, and cross-domain breadth of our study provide strong evidence for a previously overlooked phenomenon.
>
> As a reminder, the main contributions of our work are as follows: Prior studies have explored positional bias, distractor count, and attention dynamics, but none, to our knowledge, have systematically evaluated **gold context size** as an independent factor. Our setup holds the *task, answer, question*, and *distractors* constant while varying only the size of the *same* gold evidence, yielding a robust finding: **smaller gold documents are disproportionately missed and exhibit extreme positional sensitivity**, even after controlling for five major confounders. This exposes a fundamental fragility in long-context aggregation—one that appears consistently across biomedical reasoning, math, and general QA, and across 11 modern LLMs.
>
> We now address your concerns:
>
> > W1) Post-hoc analysis of entangled variables (size vs. ratio) in §4.3 does not adequately de-couple these variables.
> >
>
> > Q1) Why are you confident the determining factor is size rather than ratio?
> >
>
> **TLDR: Ratio alone cannot explain the effect—within matched-ratio bins and under large distractor-volume changes, larger gold consistently outperforms smaller gold.**
>
> We appreciate this concern and agree that it is important.
>
> We intentionally avoided framing our claim around "ratio" since doing so would require varying both distractor size and gold context size, drastically broadening the scope of our experiments. We acknowledge this limitation in the final section ("Limitations of our study") and leave it for future work. Prior research examined distractor volume extensively, whereas *gold size* has received far less attention.
>
> We agree with the reviewer that variables are entangled (and do not claim otherwise), which is why §4 explores multiple confounding factors, including size vs. ratio. Our takeaway is **not** that we have perfectly isolated a single variable—this is notoriously difficult given the complex behavior of LLMs—but that the **pattern (in terms of gold size) holds across all controls**, with varying strength but consistent direction.
>
> Specifically, in §4.3, we bucket examples by gold-to-distractor ratio so that gold and distractors occupy comparable proportions of the haystack. Within each ratio-controlled bin, *large golds consistently outperform small golds*. This suggests that absolute size contributes independently of ratio. If the reviewer believes this analysis is insufficient, we would appreciate guidance on what additional control or design would make the decoupling more convincing.
>
> To examine the statistical strength of each variable—gold size, answer token repetition, gold-to-distractor ratio, and position—we fit a multivariate logistic regression to predict correct answers across all models for each benchmark. While gold size remains a significant independent predictor of correctness, the gold-to-distractor ratio had a slight positive effect for CBB and NQ, but a slight negative effect for NM. Full statistics will be in the appendix; below are the results for NM.
>
> | Feature | Coeff | SE |
> | --- | --- | --- |
> | Intercept | 0.58 | 0.028 |
> | Lg gold (ref=sm gold) | 1.30 | 0.051 |
> | Md gold (ref=sm gold) | 0.62 | 0.023 |
> | Pos 0.20 (ref=0.0) | -0.89 | 0.034 |
> | Pos 0.40 (ref=0.0) | -0.75 | 0.034 |
> | Pos 0.60 (ref=0.0) | -0.67 | 0.034 |
> | Pos 0.80 (ref=0.0) | -0.63 | 0.034 |
> | Pos 1.00 (ref=0.0) | -0.48 | 0.034 |
> | Gold-to-Dist Ratio | -0.33 | 0.021 |
> | Answer Token Repetition | 0.00 | 0.000 |
>
> **Across all controls, gold document size remains a significant, independent predictor of performance. Ultimately, it's not the ratio that matters most; it's the strength from more gold.**
>
> Finally, Fig. 6 (Fig.7 in revision) varies distractor volume by 3x (5 → 10 → 15 distractors), thereby substantially changing the signal-to-noise ratio while leaving gold sizes unchanged. We observe that the performance ordering (large > medium > small) is stable across all settings. If the ratio were the sole driver, we would expect pattern collapses as distractor mass increases—but it is remarkably stable.
>
> Overall, we intentionally do **not** claim to support or reject a specific "ratio" hypothesis—doing so would require a significantly expanded experimental design, which we leave to future work. The current evidence shows that multiple factors interact beyond the ratio *alone*; similarly, size *alone* may not be the complete picture. We welcome suggestions for strengthening the analysis.
>
> *reply continued to the next comment*

---

> ### Author Response · Authors · 2025-11-17
>
> > W2) Conflated variables (size vs. evidence richness).
> >
>
> > Q3) Why do large contexts contain more supporting information? Could the effect be due to higher quality?
> >
>
> **TLDR: We control to make sure all variants (short, medium, and long) convey an ~equal signal. So, “large gold context” alone cannot explain the effect.**
>
> We agree that "evidence richness" is an important dimension for analysis. Longer gold documents may be higher quality—they can provide contextualization or rationalization for the answer. However, characterizing this "richness" in a formal or quantifiable way is challenging.
>
> In our approach, the goal was to maintain objective choices; we followed two principles:
>
> - **Gold-only baselines are equally strong:** We controlled for answer richness via gold-only baselines (Appendix B.2). All gold versions (small/medium/large) yield **uniformly high accuracy in isolation** when shown alone, without distractors. This demonstrates that **the smallest gold contains all the information needed to solve the task.** Therefore, differences in performance under distractors are not due to insufficient content or reasoning completeness.
> - **Larger gold documents do not repeat the answer more:** We measure answer token repetition and group tasks into comparable repetition ranges (§4.2). Within each repetition bucket, larger gold contexts still outperform smaller ones. This shows that the advantage of larger contexts is not merely due to having "more copies" of the answer—it cannot be reduced to a trivial signal-amplification explanation.
>
> We believe we have controlled for richness as thoroughly as possible within the constraints of naturally occurring datasets. If the reviewer has suggestions for how to operationalize or measure richness more directly, we would welcome them.
>
> > Q2) Mechanism of failure: attention failure or aggregation failure?
> >
>
> **TLDR: Disentangling attention from aggregation is fundamentally hard; we can run probes/attention maps, but both are correlational and cannot uniquely identify the failure mode.**
>
> **We a**ppreciate this question, and we agree it is an important one. Unfortunately, we believe these two mechanisms—*attention* and *aggregation*—are inherently difficult to disentangle in modern LLMs. The attention mechanism itself simultaneously (1) allocates focus across tokens and (2) aggregates information through value vectors; this process is repeated across many heads and layers with potentially heterogeneous behaviors. As a result, isolating “failure to attend” from “failure to integrate” is not straightforward, not only in our setting but for long-context reasoning more broadly.
>
> That said, we have considered two possible analyses:
>
> - **Probing intermediate representations:** Prior work (e.g., [1]) shows that models often encode the correct answer internally even when they fail to express it. One could apply a similar probing to our setup to test whether small-gold failures reflect “known but unused” information and track the probe scores as the gold length increases. While potentially informative, such probes are correlative and highly distribution-dependent, making it hard to draw conclusive lessons.
> - **Visualizing/analyzing attention patterns:** One could also compare attention distributions (e.g., final-layer attention mass on gold vs. distractors) between small and large gold conditions. This would indicate whether small gold contexts receive systematically less attention. However, attention inspection is also observational, and changes in attention mass do not necessarily imply causal influence on model decisions (various works have discussed this “attention vs explanation” question [2] ).
>
> We are happy to run either analysis if the reviewer believes it would add value. However, we expect both would remain correlational and would not definitively separate "attention failure" from "aggregation failure"—a fundamental challenge beyond the scope of this work. If the reviewer has suggestions for analyses that could better probe this distinction, we welcome them and would be glad to incorporate additional experiments.
>
> [1]  Insights into LLM Long-Context Failures: When Transformers Know but Don't Tell. 2024 https://arxiv.org/abs/2406.14673
>
> [2] Is Attention Explanation? An Introduction to the Debate, 2022 https://aclanthology.org/2022.acl-long.269.pdf
>
> ---
>
> **We look forward to addressing any remaining questions during the rebuttal and hope these updates warrant a higher evaluation.**

---

### Official Review · Reviewer_XMky · 2025-11-01

**Soundness:** 3
**Presentation:** 2
**Contribution:** 3
**Rating:** 6
**Confidence:** 4

**Summary:**

The paper studies how the size of the relevant passage (“needle”) affects long-context QA when mixed with many distractors (“haystack”). Results show that the larger the needle, the less impact of positional bias of relevant fact in long context (e.g., the previously found lost in the middle effect). Analysis was performed on 3 datasets (CARDBiomedBench, NaturalQuestions, and NuminaMath 1.5) and across closed (OpenAI, Gemini models) and open-source (DeepSeek, Phi-4, and Llama) models. The authors also analyze confounders (gold position, answer repetition, gold-to-distractor ratio, distractor volume, domain specificity) and claim the size effect persists. The main finding is that short “needles” are systematically missed more often.

**Strengths:**

- main finding is clear
- strong empirical evidence supporting the impact of needle size across different domains and models
- confounder checks (answer repetition, gold/distractor ratio, etc.) strengthen the core claim

**Weaknesses:**

- The proposed mitigation (balance sizes of needles, L469-470) seems oversimplified and unconfirmed. Since the gold needle isn't known a priori, we cannot enlarge only the gold; making all passages similarly long may be the only option. Because experiments vary only the gold size, it's unclear whether balancing helps in practice. Evaluating a setup where all passages (not just gold) are large (and/or of other but equal length) could clarify this.
- The single-needle setup mainly probes retrieval. Labeling results as “aggregation ability/performance” (L39-45, L228, and others) overstates scope; true aggregation requires combining multiple relevant needles. Either rephrase the claim or add multi-needle (2+ complementary/conflicting needles, multi-hop) evaluations and report if size/position effects persist.
- There are some clarity and presentation issues listed in the questions section.

**Questions:**

- If all passages (gold and distractors) are made equal in length, do positional biases and the size effect persist?
- Please clarify on “real-world system”: e.g., L166-167, L136-137: “Quantities per benchmark were calibrated to match token distributions observed in a real-world multi-agent retrieval system (∼20k tokens).” Which system(s) specifically? Please name or characterize them and provide evidence for the “~20k tokens”.
- Figure 1, axis Y, quality rate, what is it? What dataset is used? It is unclear from the figure and caption.
- Figure 5, what model is used here? It is not clear from the figure.

comments:
- Figure 7: the labels with the model names on the x-axis are difficult to align with the curves. The plot is hard to read.

---

> ### Author Response · Authors · 2025-11-18
>
> We thank the reviewer for the thoughtful and constructive evaluation. We are grateful for the positive remarks, including: *“the main finding is clear”, “strong empirical evidence”, “confounder checks … strengthen the core claim”.* We appreciate the reviewer’s recognition of both the clarity of the central finding and the methodological rigor of the analysis.
>
> As a reminder, here are the main contributions of our work: Prior work has analyzed positional bias, distractor count, and attention dynamics, but—as the reviewer notes—the relationship between **gold context size** and long-context performance has remained unexplored. Our work is, to the best of our knowledge, the **first to systematically vary the size of the *same* gold evidence while holding the task, question, and distractors fixed**, allowing us to reveal a robust and surprising finding: **smaller gold documents are disproportionately missed, and their positional sensitivity is dramatically amplified, even when controlling for five major confounders.** This exposes a fundamental fragility in realistic aggregation scenarios and provides a diagnostic lens. We also highlight cross-domain generality: biomedical, mathematical, and general-knowledge tasks all exhibit this hidden variable, with consistent patterns across eleven LLMs.
>
> We now address your comments:
>
> > W1) Mitigation suggestion seems oversimplified.
> >
>
> **TLDR: We clarified that we’re *not* suggesting enlarging gold docs but highlighting a design principle: avoid mixing extremely short and very long evidence.**
>
> We appreciate this comment and agree that our mitigation wording should be clearer. Our intention was ***not*** to suggest *enlarging the gold document* (which is unknown at inference time), but rather to point out a suggestive design principle grounded in our findings: when pipelines mix extremely short and very long retrieved passages, small but critical evidence may be systematically overshadowed. The takeaway for practitioners is that this insight can inform the design of their systems (agentic or retrieval-based systems): by ensuring that the evidence documents being mixed—**regardless of which one will ultimately contain the answer**—are **kept reasonably balanced in length**, one can reduce the likelihood that heterogeneity in document sizes would create unintended attention asymmetries.
>
> We have revised the text to clarify this. Please let us know if you’re happy with our rewording.
>
> > W1) Evaluating a setup where all passages (not just gold) are large (and/or of other but equal length) could clarify this.
> >
>
> > Q1) If all passages (gold and distractors) are made equal in length, do positional biases and the size effect persist?
> >
>
> **TLDR: Equal-length settings have been studied before; our contribution is showing that *relative* gold size matters—a dimension prior work explicitly did not vary.**
>
> The “equal-length” setup is exactly the setting that has traditionally been studied [1,2], which explicitly evaluates needle-in-a-haystack scenarios in which all passages are **uniform in length**, and indeed finds that positional effects persist when the gold/evidence passages have comparable length. But crucially, these works do **not** explore the impact of *relative gold context size*, which is our focus.
>
> Regarding varying document sizes, if all passages (gold and distractors) are made equal in length, there is prior work that examines precisely this setup. Levy et al. [3] maintain **equal-length** passages across **both gold and distractors** and then vary this length while keeping the total number of tokens in the context. They find that when the documents are longer, the gold becomes easier to retrieve—an effect complementary to ours but arising under a different control structure.
>
> [1] Lost in the Middle: How Language Models Use Long Contexts  https://arxiv.org/abs/2307.03172
>
> [2] Positional Biases Shift as Inputs Approach Context Window Limits  https://openreview.net/pdf?id=vlUk8z8LaM
>
> [3] More Documents, Same Length: Isolating the Challenge of Multiple Documents in RAG https://arxiv.org/abs/2503.04388
>
> > W2) “Aggregation ability” may overstate the scope of a single-needle setting.
> >
>
> We agree and appreciate the opportunity to refine the framing. We have revised the language throughout to *avoid* implying multi-needle aggregation. We now clearly state that our study focuses on **needle identification and selection under distractor interference**, which is a special variant of the broader “aggregation” problem.
>
> *reply continued to the next comment*

---

> ### Author Response · Authors · 2025-11-18
>
> > Q2) Please clarify on “real-world system”: e.g., L166-167.
> >
>
> **TLDR: We expanded the main text for clarity.**
>
> We apologize for the lack of clarity in the main text; these details are in Appendix A.1, under the CARDBiomedBench paragraph. As we elaborate in the appendix, this dataset includes tasks annotated by expert biomedical researchers. To compile the distractors, we developed a multi-agent LLM system that consists of independent domain-specific retrieval agents that query diverse biomedical resources, including Google search scoped to NIH domains, PubTator3.0, the Human Gene Mutation Database (HGMD), and NCBI gene/variant pages. For this specific setup, we chose these distractors to reflect realistic retrieval noise from an operational biomedical assistant system. This grounding in a real-world setting enhances the applicability of our findings.
>
> We have clarified these details in the main text and the appendix.
>
> > L136-137; Please name or characterize them and provide evidence for the “~20k tokens”.
> >
>
> The ~20k-token haystack size is best interpreted as a *design choice.* This is a deliberate design choice that provides a sufficiently long distractor context to be considered “long-context”  while remaining computationally manageable for extensive experimentation. We have reworded the text (and dropped the mention of “observed in a real-world multi-agent retrieval system”) to make this more precise. We thank you for raising this and have corrected it.
>
> > Q3) Figure 1, axis Y, quality rate, what is it? What dataset is used?
> >
>
> QualityRate in Figure 1 is a measure of response quality for CARDBiomedBench (defined in “A.4 Metrics”) averaged across 11 models. We have revised the caption to clarify and added a subsection in the main text to define the benchmark metrics.  Thank you for raising this.
>
> > Q4) Figure 5, what model is used here? It is not clear from the figure.
> >
>
> We apologize for the omission. The results here are aggregated across all models. We have clarified this in the caption.
>
> > Q5) Figure 7: the labels with the model names on the x-axis are difficult to align with the curves. The plot is hard to read.
> >
>
> The revised figure is significantly more readable. Thank you!
>
> ---
>
> **We look forward to addressing any remaining questions during the rebuttal and hope these updates warrant a higher evaluation.**

---

### Author Response · Authors · 2025-12-03
**Summary of Reviewer Comments and Changes to the Manuscript**

We thank the Area Chairs for your careful consideration of our submission and all reviewers for their thoughtful feedback. Below, we summarize the main changes and additions made in the revised manuscript.

*Note: Changes in the main paper are highlighted in blue for easy identification.*

**1. Clarified Scope, Claims, and Positioning**

- **Refined “aggregation” terminology in the text (**`XMky`**-W2, R4-W2):**

    Reworded throughout to avoid overstating the scope. We now describe the setting as “needle-in-haystack” (i.e., *needle identification and selection under distractor interference)*, rather than generic “aggregation ability,” and clearly separate our *single*-needle setting from *multi*-needle / multi-hop reasoning.

- **Clarified mitigation takeaway in the text (`hRbP`-W1):**

    Rephrased the discussion of “balancing needle sizes” to emphasize a *design principle* for system builders (avoid mixing extremely short with very long evidence documents), which was confused with a prescriptive method for enlarging the unknown gold passage.

- **Expanded discussion of novelty and contribution in the text (`9NSP`-W1, `o2Qb`-W1):**

    Strengthened the introduction,  discussion, and related work sections to more explicitly contrast our work with prior studies that assume *equal-length* passages or focus primarily on distractor count / position, highlighting that *we are the first to systematically vary the size of the **same** gold evidence* under *fixed tasks and distractors*.


### 2. Additional Analyses on Confounders and Mechanisms

- **New multivariate analysis of confounders added to the text (`hRbP`-W1, `9NSP`-W3):**

    Added a mixed-effects model (regression analysis; included in Section 4.6 and Appendix C.4) for jointly modeling the effect of multiple variables (gold size, position, gold-to-distractor ratio, and answer-token repetition). Results show that gold document size remains a strong, independent predictor of success even when controlling for these factors.

- **Expanded treatment of answer repetition vs. size (`hRbP`-W2, `hRbP`-Q3, `9NSP`-W3):**

    Added detailed tables and plots for both *exact mentions* and *repetition rate* across gold sizes and distractors, and clarified why repetition rate is a more informative metric. We explicitly show that our size effect persists within repetition-controlled buckets and cannot be reduced to trivial signal amplification.

- **Clarified interpretation of “ratio” and distractor volume (`hRbP`-W1, `hRbP`-Q1):**

    Strengthened Section 4.3 to explain how our bucketed-ratio analysis and distractor-volume sweeps (5→10→15 distractors) jointly argue that absolute gold size has an effect beyond gold-to-distractor ratio alone, while also clearly stating the limitations of not fully orthogonally varying both.

- **Our scope is broader than RAG (`9NSP`, `o2Qb`):** We explicitly clarified (in the text and our responses) that our findings apply to *any agent system that mixes heterogeneous evidence*.  We now emphasize this generality in the Introduction and Discussion.

### 3. Figure, Caption, and Presentation Improvements

- **Clarified figure captions and model identities (**`XMky`**-Q3,** `XMky`**-Q4):**

    Updated captions for Figure 1 (definition of QualityRate, dataset, and averaging protocol) and Figure 5 (explicitly stating that results are aggregated across all models).

- **Improved figure readability and labeling (**`XMky`**-Q5, `9NSP`-W4):**

    Redesigned figures with rotated / resized x-axis labels, clearer color/marker legends, and better alignment between curves and model names, especially for the per-model plots (Figures 3 and 7 in the previous version).


### 4. Remaining comment — Mechanistic analysis expectations:

We also respond directly to the suggestion for mechanistic interpretability. While we agree with the aspiration of providing causal explanation of model behavior, *this goal is itself an active research area* and lies beyond the scope of our stated goal. Existing methods (attention maps, value-flow tracing, probing, activation patching) are largely correlational, not causal.  *We invited the reviewer to specify what concrete mechanistic experiment they had in mind, but no actionable proposal was provided.*

In our view, developing causal mechanistic accounts is a natural and valuable direction *building on top of* our empirical findings, and we have clarified in the Discussion that we see this as important future work rather than a requirement for the present contribution.

---

### Meta-Review · Area_Chair_Yt14 · 2026-01-06

**Summary:**

This paper presents a large-scale empirical study showing that LLMs are significantly more likely to miss relevant information when the gold context is short, especially when it appears later in a long input. Reviewers broadly agreed that the experiments are extensive, carefully controlled, and consistently demonstrate a strong interaction between gold context size and positional bias across models and domains. However, despite the robustness of the empirical findings, the consensus was that the contribution is primarily observational and incremental relative to existing works. Key concerns remained about disentangling gold size from related factors (signal-to-noise ratio and evidence richness), limited mechanistic insight, and insufficient validation in more realistic end-to-end or multi-needle aggregation settings. As a result, the paper was judged to fall short of the bar for acceptance.

**Reviewer Concerns:**

While some presentation issues (figures, captions, metric definitions, and wording around "aggregation") were clarified, and the additional experiments were helpful to support the empirical evidence in the rebuttal, there are remaining concerns as follows:

- Lack of mechanistic explanation (attention vs. aggregation failure) persisted; reviewers felt even partial or lightweight mechanistic evidence would substantially strengthen the paper.
- External validity concerns remained, especially the absence of convincing end-to-end RAG or multi-needle/realistic aggregation evaluations demonstrating the effect under full pipelines.
- Overall novelty and impact were still viewed as modest: the findings are intuitive and well-supported, but not sufficiently explanatory or system-level to meet the conference bar.

**Reviewer Scores:**

After reading the rebuttal, Reviewer o2Qb explicitly mentioned that the original score is kept. Given the major remaining concerns summarized above, other reviewers are unlikely to have revised their original scores.

---

### Decision · Program_Chairs · 2026-01-26

Reject